# Cellular and biophysical barriers to lipid nanoparticle mediated delivery of RNA to the cytosol

Johanna M. Johansson [1,7], Hampus Du Rietz [1,7], Hampus Hedlund [1,7], Hanna C. Eriksson [1], Erik Oude Blenke [2], Aditya Pote [3], Said Harun [2], Pontus Nordenfelt [4], Lennart Lindfors[2] & Anders Wittrup [1,5,6] ✉

Lipid nanoparticles (LNPs) are clinically approved for mRNA-based vaccines and liver-targeted siRNA delivery. However, poor nucleic acid delivery efficiency limits their application in extrahepatic tissues and tumors. Here, using live-cell and super-resolution microscopy, we identify multiple distinct steps of inefficiencies in the cytosolic delivery of both siRNA and mRNA cargoes. Membrane damages marked by galectin recruitment are conducive to cytosolic RNA release, whereas membrane perturbations recruiting the ESCRT machinery do not permit endosomal escape. Notably, only a small fraction of RNA is released from galectin-marked endosomes and, unexpectedly, many damaged endosomes contain no detectable RNA cargo. Using LNPs with both fluorescently labeled ionizable lipid and RNA, we show that these components segregate during endosomal sorting – both within single endosomes and across endosomal compartments. Finally, we visualize localized ionizable lipid enrichment in endosomal membranes and membrane damage in direct proximity to siRNA-LNPs tethered to luminal vesicle membranes. Taken together, our findings reveal multiple mechanistic barriers limiting intracellular RNA delivery by LNPs.

Several different nucleic acid-based therapeutics, including antisense[1], siRNA[2,3], and mRNA-compounds[4,5] have recently entered clinical use in diverse clinical settings. While cytosolic delivery using non-particulate formulations has proven feasible for relatively small antisense and siRNA molecules, larger molecules such as mRNAs or CRISPR-based compounds[6,7] currently require formulation in lipid nanoparticles (LNPs) for functional delivery. Despite substantial recent efforts to modify LNP compositions to target tissues beyond the liver[8–15], current LNPs are severely limited in both tissue tropism and delivery efficiency. Limited escape out of endosomes into the cytosol of target cells is generally considered to be the rate limiting step for RNA delivery[16–18].

Still, little detail is known about the cellular, biophysical and molecular barriers to efficient endosomal escape[16]. Consequently, rational strategies to overcome these barriers are difficult to devise and most efforts to improve current LNPs focus on high-throughput screening strategies with limited basis in mechanistic understanding.

LNPs are formulated with multiple lipophilic components with empirically determined ratios, including a key ionizable lipid with significant ionization around the pH of early endosomes (pH 6.0–6.5)[19]. For example, in the first FDA- and EMA-approved LNP formulated therapeutic, patisiran[2], the mid-point of the ionization vs. pH curve[20] for the ionizable lipid DLin-MC3-DMA (usually referred to as

[1]Lund University, Department of Clinical Sciences Lund, Section for Oncology, Lund, Sweden. [2]Advanced Drug Delivery, Pharmaceutical Sciences, Bio-Pharmaceuticals R&D, AstraZeneca, Gothenburg, Sweden. [3]Advanced Drug Delivery, Pharmaceutical Sciences, BioPharmaceuticals R&D, AstraZeneca, Waltham, MA, USA. [4]Lund University, Department of Clinical Sciences Lund, Section for Infectious Medicine, Lund, Sweden. [5]Skåne University Hospital, Lund, Sweden. [6]Wallenberg Center for Molecular Medicine, Lund, Sweden. [7]These authors contributed equally: Johanna M. Johansson, Hampus Du Rietz, Hampus Hedlund. ✉e-mail: anders.wittrup@med.lu.se

the $pK_a$) is reported to be 6.44[19]. It has been proposed that upon entering the acidifying endosomal system the ionizable lipid of the LNP becomes protonated and interacts with negatively charged endosomal luminal membrane lipids[21]. This interaction is believed to result in a phase shift of the planar limiting endosomal membrane to an endosomal escape conducive inverted hexagonal membrane lipid phase[22]. However, how the transition to an inverted hexagonal phase results in endosomal escape is not well understood. Through the use of synthetic lipid membranes, it has been shown that lowering the pH triggers tight interaction of LNPs and the synthetic membrane as well as lipid exchange between the LNP and the membrane[23]. This tight interaction and lipid transfer could possibly be a trigger of endosomal damage. Indeed, in cellulo, LNPs trigger membrane disruptions detectable with membrane damage sensing galectins and these damages are associated to the cytosolic release of siRNA[24]. The induction of galectin-positive membrane damages is also statistically correlated to productive mRNA delivery in different delivery formulations[25,26]. Still, it is not known if the cytosolic release of the large mRNA molecules also proceeds from vesicles exhibiting galectin recruitment. Other pathways of delivery are conceivable, and recently the Endosomal Sorting Complexes Required for Transport (ESCRT) machinery was demonstrated to be involved in sensing and repairing endosomal membrane damages[27,28]. It has also been proposed that a complex salt of RNA and ionizable lipid can be transferred from the LNP into the vesicle membrane, whereafter the RNA could translocate to the cytosolic side without forming a pore[29,30].

Here, we set out to characterize the LNP-mediated cytosolic delivery of both siRNA and mRNA using MC3-based LNPs. Using live-cell imaging and super-resolution microscopy of LNPs with fluorescently labeled RNA payload, we identify multiple distinct steps of substantial inefficiency in the cytosolic delivery of nucleic acid cargoes. We show that membrane damages triggering galectin recruitment are conducive to both siRNA and mRNA release but only a subset of internalized LNPs trigger such damages. By analysis of individual endosomes, we also show that only a small fraction of the nucleic acid cargo contained in the endosome is released to the cytosol upon LNP-triggered membrane damage. Unexpectedly, many damaged endosomes contain no detectable nucleic acid cargo. By using LNPs containing both fluorescently labeled ionizable lipid (BODIPY-MC3) and siRNA, we show that the RNA payload and ionizable lipid segregate during endosomal sorting. Finally, we visualize disintegration of LNPs inside the lumen of endosomes and membrane damages in direct proximity to LNPs tethered to the endosomal membrane. Taken together our work has identified multiple mechanistically different barriers to efficient LNP delivery. The methods presented here will enable rational strategies to overcome these barriers and make LNP mediated RNA delivery feasible in additional tissues and new clinical settings.

## Results

### Endosomes damaged by RNA-LNPs frequently lack significant RNA payload

We first set out to characterize endosomal membrane damages induced by MC3-LNPs. While multiple members of the galectin family can be used to detect membrane damages[31], galectin-9 is the most sensitive membrane damage sensor when damages are induced by small molecule drugs[32] or transfection lipids[24]. In addition, during transfection lipid-mediated siRNA delivery, the releasing endosome is detected with close to 100% sensitivity using galectin-9[33]. Accordingly, for both siRNA- and mRNA-LNPs, uptake was fast and triggered recruitment of galectin-9 to vesicular structures within 1 h as evidence of endosomal damage (Fig. 1a and Supplementary Movie 1). The uptake of RNA-LNPs and the number of galectin-9 foci was dose-dependent and reached a plateau above 50 nM (0.72 μg mL$^{-1}$) for siRNA- and 0.75 μg mL$^{-1}$ for mRNA-LNPs (Supplementary Fig. 1a and Fig. 1b), where

the rate of LNP internalization appeared saturated (Supplementary Fig. 1b). siRNA- and mRNA-LNP uptake strongly correlated with galectin-9 recruitment (Supplementary Fig. 1c), however, only a fraction of internalized LNPs triggered galectin-9 recruitment. Cellular uptake of siRNA-LNPs was faster than with mRNA-LNPs, likely contributing to the earlier induction of endosomal damages and galectin-9 response. Using fast live-cell microscopy, individual events with de novo recruitment of galectin-9 to endosomes containing LNPs could be identified (Fig. 1c). Single vesicle tracking and fluorescence quantification showed that, in endosomes before damage and galectin recruitment, the RNA content was highly heterogenous (Fig. 1d). Higher siRNA-LNP concentrations in the medium resulted in higher siRNA load in the damaged vesicles. The magnitude of galectin-9 recruitment also varied, ranging from faint to distinct and strong response (Supplementary Fig. 1d). Both siRNA- and mRNA-LNPs exhibited dose-dependent biological activity, confirming successful cytosolic delivery (Supplementary Fig. 1e). Taken together, both siRNA- and mRNA-LNPs trigger galectin-9$^+$ endosomal damages as a potential conduit to functional cytosolic delivery of LNP-formulated RNAs.

Interestingly, not all endosomes showing de novo recruitment of galectin-9, triggered by LNP treatment, contained detectable RNA payload (Supplementary Fig. 2a). With siRNA-LNP, the fraction of damaged vesicles with detectable siRNA cargo (which we refer to as the hit rate) ranged from 67% to 74% for all evaluated concentrations (Fig. 1e). On the other hand, with mRNA-LNP, the hit rate was considerably lower with most damaged vesicles having no detectable mRNA content, with a corresponding hit rate of approximately 20%. Microscopy image acquisition and analysis parameters were identical for siRNA- and mRNA-LNPs, but other technical explanations for the lower hit rate and difference between siRNA and mRNA are conceivable. To further investigate this, we first confirmed that we were indeed able to detect both single intact siRNA-LNPs and mRNA-LNPs with our standard experimental imaging setup (Supplementary Fig. 2b). To evaluate the effect of difference in fluorophore labeling (AlexaFluor 647 for siRNA and Cy5 for mRNA), we incubated cells with Cy5-labeled siRNA-LNPs, still resulting in the higher siRNA hit rate of approximately 70% (Fig. 1e). Thus, the lower hit rate of mRNA-LNPs compared to siRNA-LNPs is not dependent on the specific fluorophore used. Furthermore, we confirmed that the vast majority of intact mRNA-LNPs contained mRNA, verifying that the low hit rate was not a consequence of LNPs lacking RNA payload (Supplementary Fig. 2c).

Differences in fluorescence intensity between individual siRNA- and mRNA-LNPs, depending on the number of fluorophores per RNA and total RNAs per LNP, could impact the sensitivity of our assay, conceivably affecting the observed hit rates. Reference measurements showed that, with both intact LNPs and LNPs disrupted by Triton X-100 (TX-100), the mRNA fluorescence intensity was significantly lower than with siRNA (Supplementary Fig. 2d). With intact LNPs, siRNA-LNP intensity was ~2 times higher. After LNP disintegration, the siRNA signal increased approximately 2.6-fold, while the mRNA signal increased by less than 20%. The substantial signal increase from disintegrated siRNA-LNPs implies a high degree of fluorophore quenching in intact siRNA- but not mRNA-LNPs. We hypothesized that LNPs were likely at least partially disintegrated at the time of membrane damage, why from this data, it is reasonable to expect a ~2–5-fold difference in the siRNA- and mRNA-LNP fluorescence per nanoparticle around galectin-9 recruitment in live-cell experiments.

To assess if this difference in signal intensity could explain the lower hit rate observed for mRNA-LNPs, we also produced siRNA-LNPs where only 25% of siRNAs were labeled with AF647 (AF647¼-siRNA-LNP). We confirmed that equivalent doses of Cy5-mRNA-LNP and AF647¼-siRNA-LNP had similar fluorescence intensity both while intact and disrupted (within 2% and 40%, respectively) (Supplementary

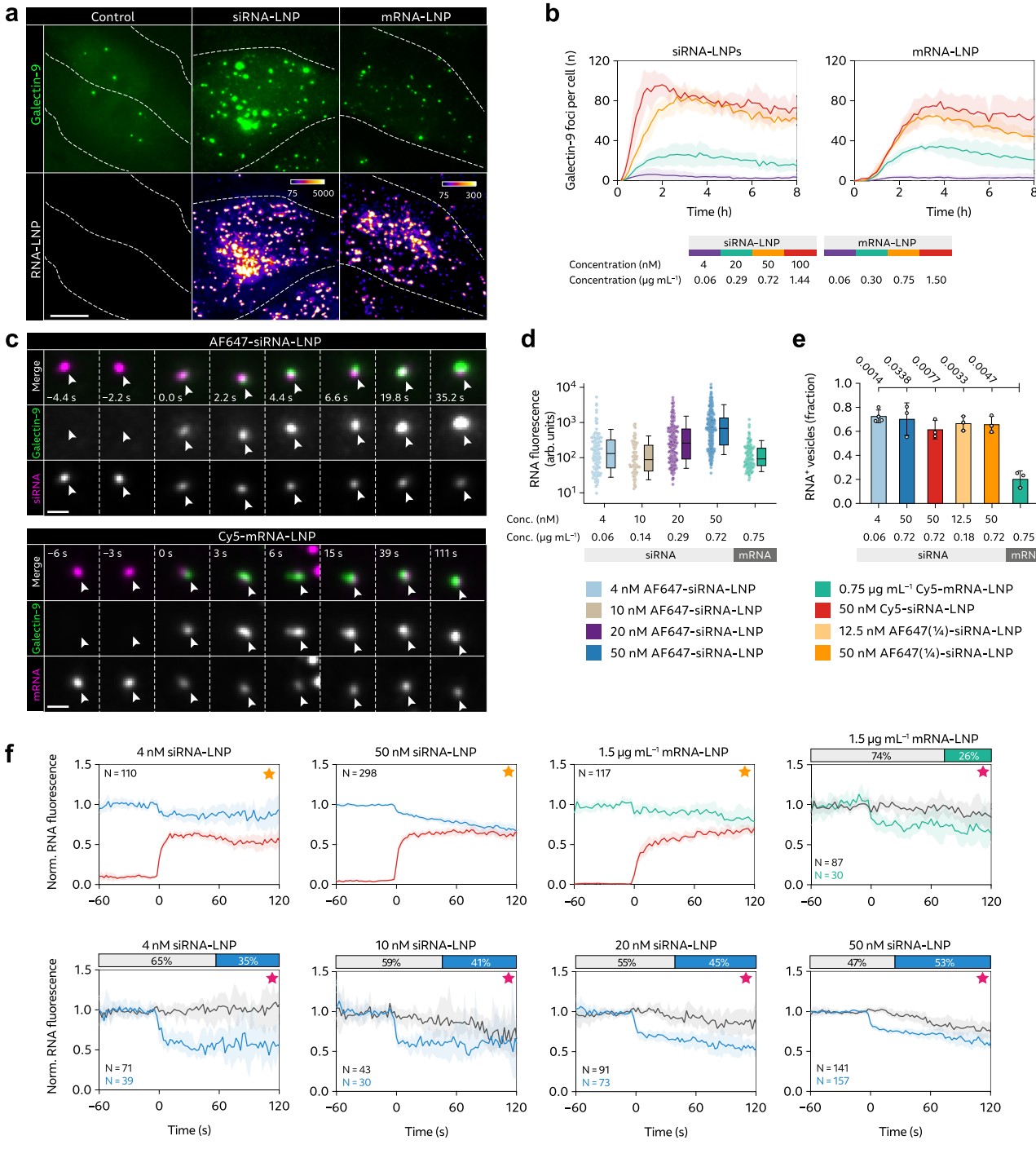

**Fig. 1 | LNP-mediated endosomal release of siRNA and mRNA is incomplete and inefficient.** HeLa cells expressing YFP-Galectin-9 (HeLa-Gal9-YFP) were incubated with fluorescently labeled siRNA- or mRNA-LNPs during live-cell widefield microscopy. **a** Representative images showing galectin-9 response and LNP uptake after 100 min continuous incubation with 100 nM siRNA-LNP or 1.5 μg mL⁻¹ mRNA-LNP. Brightness and contrast were adjusted separately. Outlines indicate cell boundaries; color bar shows intensity value range. Images are representative of 106 and 102 cells from two independent experiments. Scale bar is 10 μm. **b** Number of galecin-9 foci per cell evaluated during 8 h LNP incubation. Line is mean, shade is s.d. *N* = 4 technical replicates, see Methods for *N* details. **c** Representative images showing de novo recruitment of galectin-9 to a vesicle containing siRNA-LNP (top) and mRNA-LNP (bottom), indicated by arrowheads. Time = 0 is the first frame with detectable galectin recruitment. Scale bar is 1 μm. **d** Fluorescence intensity of vesicles containing AF647-siRNA-LNP or Cy5-mRNA-LNP before detectable membrane damage. Circles are vesicles pooled from 2–4 independent experiments per condition. Number of vesicles (N_v) and cells (N_c) analyzed were (N_v;N_c) = (115;81), (84;61), (170;82), (315;111) and (117;87) for 4, 10, 20, 50 nM siRNA-LNP and 0.75 μg mL⁻¹ mRNA-LNPs, respectively. Boxes are median ± i.q.r.; whiskers are 10–90 percentiles. **e** Fraction of all damaged vesicles with detectable siRNA or mRNA payload. Bars are mean, error bars are s.d. *N* = 3 independent experiments. Statistics: Brown-Forsythe and Welch ANOVA with two-tailed Dunnett's T3 multiple comparisons test. **f** Fluorescence intensity of individual RNA⁺ vesicles around the time of membrane damage (*t* = 0). Traces were sub-grouped to identify events with fast initial RNA release. Yellow star: all events. Magenta star: fast releasing vesicles vs. other vesicles. Lines are median, shades are 95% CI of median. Bars (top) show fraction of fast releasing vesicles vs. other vesicles. Data are pooled from 2–4 independent experiments per condition.

Fig. 2d). Evaluating the AF647¼-siRNA-LNP at two concentrations, we found that the hit rate seen with the other siRNA-LNP formulations was maintained (Fig. 1e). We also confirmed that the hit rate with both AF647-siRNA-LNP and Cy5-mRNA-LNP was consistent at different time points after LNP addition (Supplementary Fig. 3a, b) and throughout image acquisitions (Supplementary Fig. 3c), indicating no significant loss of detection sensitivity by photobleaching or potential imaging-related effects. Galectin-9 recruitment to mRNA⁻ vesicles occurred with identical kinetics as to mRNA⁺ endosomes, but the magnitude of galectin-9 recruitment was higher with the latter (Supplementary Fig. 4a). In summary, we observe that a significant fraction of LNP-induced endosomal damages occur in vesicles with no detectable RNA payload, and specifically we observe a lower hit rate for mRNA-LNPs than with siRNA-LNPs.

### LNP-mediated endosomal release of siRNA and mRNA is incomplete and inefficient

We next sought to quantify the escape of RNA cargo from individual damaged endosomes. For this, LNP-containing endosomes were tracked before and after detectable galectin-9 recruitment, measuring the RNA fluorescence intensity. For siRNA-LNPs, the fluorescence intensity was initially stable, but just as galectin recruitment was detectable, a sudden decrease of siRNA intensity was observed (Fig. 1f). With mRNA-LNP, an initial fast signal decrease at the time of galectin recruitment was less pronounced but suggestive. As the kinetics and efficiency of RNA release potentially vary between the individual damage events (given the heterogenous RNA load and galectin recruitment), the data was sub-grouped with respect to events exhibiting a sharp drop in RNA fluorescence, or not, at the time of galectin recruitment. With mRNA-LNPs, we found that 26% of all vesicles showed rapid RNA release, where the RNA content (fluorescence) decreased by 24% (median; 10–32% 95% CI of median) at -10 s after damage, and then appeared largely stable. With siRNA-LNPs, there was a higher fraction of endosomes showing fast initial release, ranging from 35% (4 nM) to 53% (50 nM), in a dose-dependent manner. The relative magnitude of the fast release was comparable to that of mRNA, between 37% with 4 nM (27–45%) and 23% with 50 nM (18–25%; median and 95% CI of median) at -10 s after start of galectin recruitment. The relative magnitude of siRNA release was dose-dependent, with higher release efficiency with lower siRNA-LNP concentrations. At the LNP dose most relevant in vivo (4 nM siRNA), vesicles that did not show fast initial siRNA release within the first moments after membrane damage maintained the pre-damage RNA payload throughout the observation period of -2 min. Still, a slow or gradual endosomal escape mechanism cannot be completely ruled out, especially at higher (supra-physiological) doses. Alternatively, the gradual drop seen at higher doses could also constitute non-synchronized additional rapid release events from vesicles loaded with multiple LNPs. Taken together, we see an incomplete (<50%) but fast (<10 s duration) release of both siRNA and mRNA, but only from a subset of all damaged endosomes containing RNA payload.

Separating LNP⁺ endosomes with low, medium or high RNA fluorescence intensity revealed no clear difference in the kinetics of siRNA release, but a higher relative release efficiency of vesicles with lower RNA payload (Supplementary Fig. 4b). This confirms that the LNP load per vesicle contributes to the difference in release magnitude observed depending on siRNA-LNP concentration. No difference in mRNA release depending on the RNA load was observed. When subgrouping events with respect to galectin-9 response, there was no difference in the RNA-LNP release kinetics between vesicles showing weak or strong galectin-9 recruitment (Supplementary Fig. 4c). In summary, this suggests that the total RNA load (or number of LNPs) in damaged vesicle does not affect the release kinetics, and only marginally influences the release efficiency within the observed period.

### ESCRT recruitment limits RNA-LNP induced membrane damage

The inefficient RNA release from damaged endosomes marked by galectin-9 made us ask if there could be alternative membrane perturbations, that could be more efficient in permitting endosomal release of LNP-RNA. It has been shown that components from the Endosomal Sorting Complex Required for Transport (ESCRT) are recruited to damaged endolysosomes, initiating membrane repair[27,28]. Depending on the nature of membrane damage, ESCRT components could act alone or with associated galectin recruitment. To investigate the role of ESCRTs in LNP mediated RNA delivery, cells expressing GFP-CHMP2A were incubated with siRNA- or mRNA-LNPs during live-cell microscopy. After -1–3 h of LNP internalization, we observed formation of new and increasing numbers of CHMP2A foci (Fig. 2a–c and Supplementary Movie 2), most of which colocalized with endosomes containing RNA (Fig. 2d). With mRNA-LNPs, CHMP2A response was also evident but less pronounced compared to siRNA-LNP, which at least in part could be explained by slower internalization and intracellular accumulation shown earlier.

To determine if membrane perturbations detected by CHMP2A, but not galectins, could permit endosomal release of LNP-siRNA, we used fast time-lapse microscopy to follow de novo recruitment of GFP-CHMP2A to vesicles containing siRNA-LNPs. As expected, a fraction of events detected by CHMP2A also displayed galectin recruitment, as evaluated simultaneously with mRuby-galectin-8, and were not included in the further analysis. Single-vesicle measurements of siRNA fluorescence intensity in CHMP2A⁺ but galectin-8⁻ vesicles did not show evidence of endosomal release around the time of CHMP2A recruitment (Fig. 2e, f). Taken together, this implies that a subset of membrane perturbations triggered by siRNA- and mRNA-LNPs are detected by the ESCRT machinery but not galectins during the observable period. However, this subset of damages does not permit release of siRNA payload.

Depletion of ESCRT components have previously been shown to compromise endolysosomal membrane damage repair[27,28,34]. By maintaining endosomal membrane integrity, ESCRT-mediated membrane repair could potentially reduce the rate at which LNPs trigger membrane damages recruiting galectins, thereby counteracting cytosolic RNA delivery and limiting the overall efficiency. To investigate this, we inhibited ESCRT activity using siRNA-mediated knockdown of three key components involved in the membrane repair response, namely Alix, Tsg101 (ESCRT-I subunit)[27] and CHMP6 (ESCRT-III subunit)[34]. Target knockdown was confirmed on the mRNA level 24 h after siRNA treatment using reverse-transcription quantitative PCR (RT-qPCR) (Supplementary Fig. 5a). The LNP uptake was not affected by ESCRT knockdown (Supplementary Fig. 5b). When incubated with mRNA-LNPs, single knockdown of CHMP6 or dual knockdown of Alix and Tsg101 did not alter the number of newly formed galectin-9 foci compared to control cells (Fig. 2g and Supplementary Fig. 5c). In contrast, dual knockdown of CHMP6 and Tsg101 and simultaneous knockdown of CHMP6, Tsg101 and Alix both resulted in ~20% more foci per cell at the peak around 4–8 h after start of LNP treatment (Fig. 2g, h). This suggests that the observed ESCRT response to LNP-mediated membrane perturbations plays a role in limiting the formation of galectin-9⁺ membrane damages, that could otherwise allow additional endosomal release of RNA payload.

### LNP-mediated release of RNA payload occurs from Rab5⁺ EEA1⁺/⁻ early endosomes

After finding that galectin-9 could indeed detect productive but incomplete release events of both siRNA- and mRNA-LNPs, we next asked what endosomal compartments are damaged by the LNP. Transfection lipid-mediated siRNA release occurs from early to maturing Rab5⁺ and Rab7⁺ endosomes[24] while membrane destabilizing cationic amphiphilic compounds can mediate RNA delivery from late endosomes and lysosomes[32]. To probe this aspect of LNP delivery, we first observed that a fraction of siRNA-LNPs were initially localized to

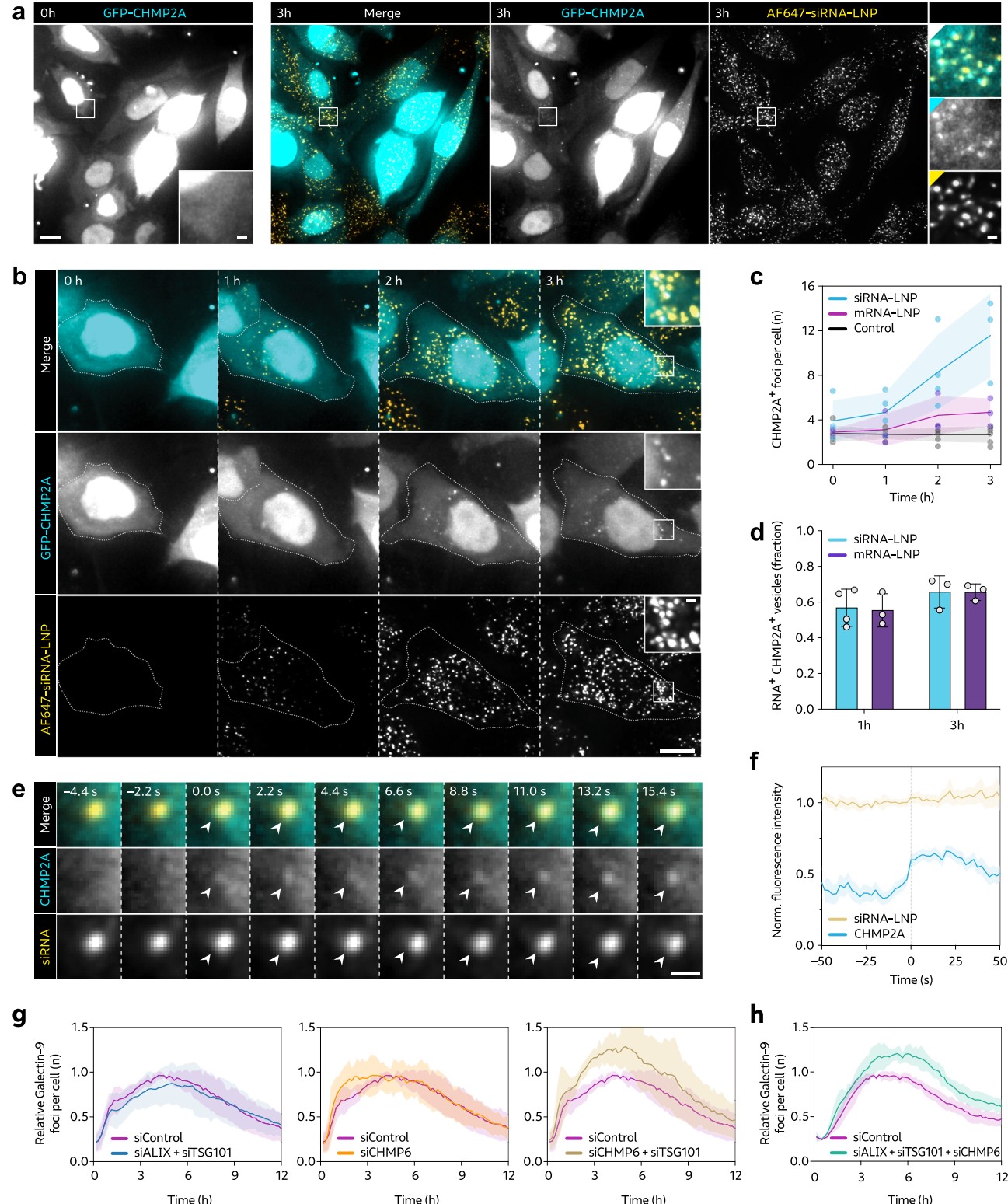

Rab5+ early endosomes followed later on by ~50% of the internalized LNPs localizing to CD63+ late endosomes for the remaining period of the observation (Supplementary Fig. 6a), consistent with typical endosomal cargo trafficking. We also observed that the abundance of both early (Rab5+) and late (CD63+) endosomes did not change during LNP incubation (Supplementary Fig. 6b),

Next, we investigated various endosomal compartments (early, maturing/late and recycling endosomes and lysosomes), together with galectin-9 during high-speed live-cell microscopy. We measured

endosomal marker abundance on vesicles with de novo recruitment of galectin-9 to during siRNA- or mRNA-LNP incubation (Fig. 3a, b). Primarily early endosomal compartment markers were present on damaged endosomes and Rab5 was the most frequent marker of damaged vesicles, present on 62% and 94% of vesicles containing siRNA- or mRNA-LNPs, respectively (Fig. 3c). EEA1 was also commonly seen (28% and 63% for siRNA and mRNA, respectively), while markers of later endosomal compartments and lysosomes were rare. This result was corroborated by independent manual evaluation of microscopy

**Fig. 2 | RNA-LNP induce CHMP2A recruitment to non-releasing vesicles.** HeLa cells transiently expressing GFP-CHMP2A were incubated with 100 nM AF647-siRNA-LNP or 1.5 µg mL$^{-1}$ Cy5-mRNA-LNP, acquiring images every 30 min using a widefield microscope. **a, b** Representative images demonstrating formation of CHMP2A foci during AF647-siRNA-LNP treatment. **a** Brightness and contrast were adjusted separately for full images and insets. Scale bar, 10 µm; detail, 1 µm. **c** The number of CHMP2A foci per cell was quantified through blinded manual inspection. Lines are mean, shades are 95% CI of mean. **d** The fraction CHMP2A foci colocalizing with RNA was determined manually through visual assessment. Mean ± s.d. is shown. **a–d** Images and data are from $N = 3$, 4 and 6 independent experiments with 49, 82 or 97 cells analyzed, for Cy5-mRNA-LNP, AF647-siRNA-LNP and untreated control cells, respectively. **e, f** HeLa cells expressing mRuby-galectin-8 and GFP-CHMP2A were incubated with 50 nM AF647-siRNA-LNPs and imaged using widefield microscopy. siRNA$^+$ but galectin-8$^-$ vesicles showing recruitment of CHMP2A were identified and tracked in 4D. **e** Representative image details showing CHMP2A recruitment to a siRNA$^+$ endosome, indicated by arrows. Scale bar is 1 µm. **f** CHMP2A and siRNA fluorescence intensity of tracked endosomes. Lines are mean, shades are 95% CI of mean. Traces are aligned in time so that CHMP2A recruitment was first detected at $t = 0$. $N = 118$ vesicles, with 79 cells analyzed, from 3 independent experiments. **g, h** HeLa-Gal9-YFP cells were transfected with siRNA targeting ESCRT components Alix, Tsg101 or CHMP6, or negative control siRNA, in the combinations detailed in the plots. Cells were continuously incubated with 0.5 µg mL$^{-1}$ mRNA-LNP, while evaluating the galectin-9 response using confocal microscopy. Mean ± s.d. from three independent experiments are shown, with at least 118 (**g**) or 149 (**h**) cells per timepoint for all conditions and experiments.

images, where Rab5 again was the most prevalent endosomal marker followed by EEA1 (Supplementary Fig. 6c). EEA1 intensity generally peaked before endosome damage, while Rab5 was more consistent over the observed interval (Fig. 3d, e and Supplementary Movie 3). Overall, there was no significant difference in the RNA load of LNP-damaged vesicles in cells expressing EEA1, Rab5 or Rab7 (Supplementary Fig. 6d), although endosomes with presence of the compartment markers typically showed higher RNA content than vesicles classified as marker negative (Supplementary Fig. 6e). When investigating endosomal release from EEA1 and Rab5, we observed that LNPs triggered both productive and unproductive damages in these compartments (Supplementary Fig. 7a–c). The probability of a damage event triggering a fast release was slightly higher in non-marker positive structures for siRNA-LNPs, while, since virtually all damaged vesicles were Rab5$^+$ for mRNA-LNPs, the vast majority of productive release events were also Rab5$^+$. However, for siRNA-LNPs release from other endosomal compartments can also occur. In summary, we identify Rab5$^+$ EEA1$^{+/-}$ early endosomes (or very early maturing endosomes) as the primary compartment where LNPs trigger membrane damage and as the site for a significant fraction of the productive release of RNA payload.

## RNA-LNPs disintegrate within the endosomal lumen during endosomal maturation

After determining that LNP mediated membrane damage, primarily occurs in early endosomes, we next aimed to study the mechanism of LNP-triggered membrane disruption in this and related compartments. Integration of ionizable lipid into the endosomal membrane is suggested to induce membrane damage and subsequent RNA release[35]. Given this, LNPs would need to disintegrate (at least partially) for MC3 to be incorporated into the membrane. To explore this, we investigated AF647-siRNA-LNPs and Cy5-mRNA-LNPs internalized by HeLa cells expressing GFP-labeled EEA1 or the late endosome/multivesicular body (MVB) marker CD63, to determine if there was a difference in LNP appearance between earlier and later endosomal compartments. With super-resolution Airyscan microscopy, we could observe both single and multiple distinct LNP foci within the lumen of EEA1$^+$ early endosomes, often localized in direct proximity to the endosomal membrane, for both siRNA- and mRNA-LNPs (Fig. 4a, c, d). By measuring the full-width at half maximum (FWHM) of intraluminal siRNA-LNPs and comparing with reference FWHM measurements of single LNPs on glass slides, we concluded that the siRNA-LNP foci visualized in early endosomes were single LNPs, or in some cases multiple separated LNPs within the same endosome (Supplementary Fig. 8a, b). On the contrary, in CD63$^+$ MVBs, the siRNA signal was generally homogenously dispersed and filling the entire lumen, suggesting disintegration of intralumenal siRNA-LNPs (Fig. 4a, c). Using fast live-cell iSIM to acquire super-resolution $z$-stacks, LNP-siRNA could be visualized in three dimensions as particulate structures, tethered to the vesicle membrane of early endosomes (Fig. 4b and Supplementary Fig. 8c), while disintegrated LNP-siRNA in late endosomes displayed

homogenously dispersed siRNA signal within the endosomal lumen (Fig. 4b, Supplementary Fig. 8d, e and Supplementary Movie 4).

Interestingly, in Rab5$^+$ endosomes, siRNA-LNPs with varying appearance could be identified, ranging from well-defined and seemingly intact LNPs, to vesicles with partially condensed and partially luminal siRNA distribution, to homogenous siRNA distribution with only a subtle remaining siRNA focus, resembling the highly disintegrated LNPs observed in CD63$^+$ endosomes (Fig. 4c). This suggests that LNP disintegration proceeds in Rab5$^+$ early endosomes.

With mRNA-LNPs, the mRNA payload was typically more dispersed in CD63+ vesicles than in EEA1$^+$ endosomes, but with a partly condensed mRNA-LNP remnant, that was generally not observed with siRNA-LNP at this stage (Fig. 4d). A manual classification of siRNA- and mRNA-LNP appearance in EEA1$^+$, Rab5$^+$ or CD63$^+$ compartments confirmed that the majority of both siRNA- and mRNA-LNP in EEA1$^+$ endosomes appeared well-confined and tethered to the luminal membrane, while there were slightly more endosomes with dispersed siRNA signal in Rab5$^+$ endosomes (Fig. 4e). In MVBs, the siRNA was generally dispersed within the lumen, while the mRNA frequently remained confined to particulate foci near the endosomal membrane. To confirm that the observed difference between siRNA- and mRNA-LNPs disintegration was not caused by difference in fluorophore labeling, GFP-CD63 expressing cells were treated with Cy5-labeled siRNA-LNPs. Cy5-siRNA likewise exhibited homogenously dispersed luminal signal (Supplementary Fig. 8f), indicating that the variation in LNP disintegration was a consequence of the difference in RNA cargo. Furthermore, we noticed a more pronounced increase in far-red fluorescence of AF647-siRNA LNPs compared to Cy5-mRNA LNPs during endosomal maturation (Supplementary Fig. 8g). Dissociation of intact AF647-siRNA LNPs with TX-100 showed ~3-fold AF647 intensity increase, due to dequenching (Supplementary Fig. 2d). However, intact Cy5-mRNA LNPs were not significantly quenched, indicating that the signal increase observed from siRNA-LNPs in late endosomes was caused by dequenching of AF647, further supporting that LNP-RNA disintegrates within endosomes during endosomal maturation.

## Ionizable lipid and siRNA cargo separate within single endosomes and to different endosomal compartments

As shown above, a significant fraction of disrupted endosomes did not contain RNA cargo. This made us consider if such RNA-depleted endosomes are damaged by the presence of the ionizable lipid alone, because of separation of the lipid and RNA into different endosomal compartments. We also observe that LNPs dissociate within the endosomal lumen, consistent with earlier whole-cell FRET measurements of dual-labeled siRNA formulated in LNPs suggesting rapid dissolution of LNPs[36]. Thus, we next aimed to study LNP disintegration and separation of ionizable lipid and RNA payload, within single endosomes. To this end, we designed LNPs incorporating both fluorescently labeled siRNA (AF647) or mRNA (Cy5) and a fluorescent derivative of the MC3 lipid with one lipid tail substituted with BODIPY (BODIPY-MC3). We evaluated the performance of BODIPY-MC3 LNPs

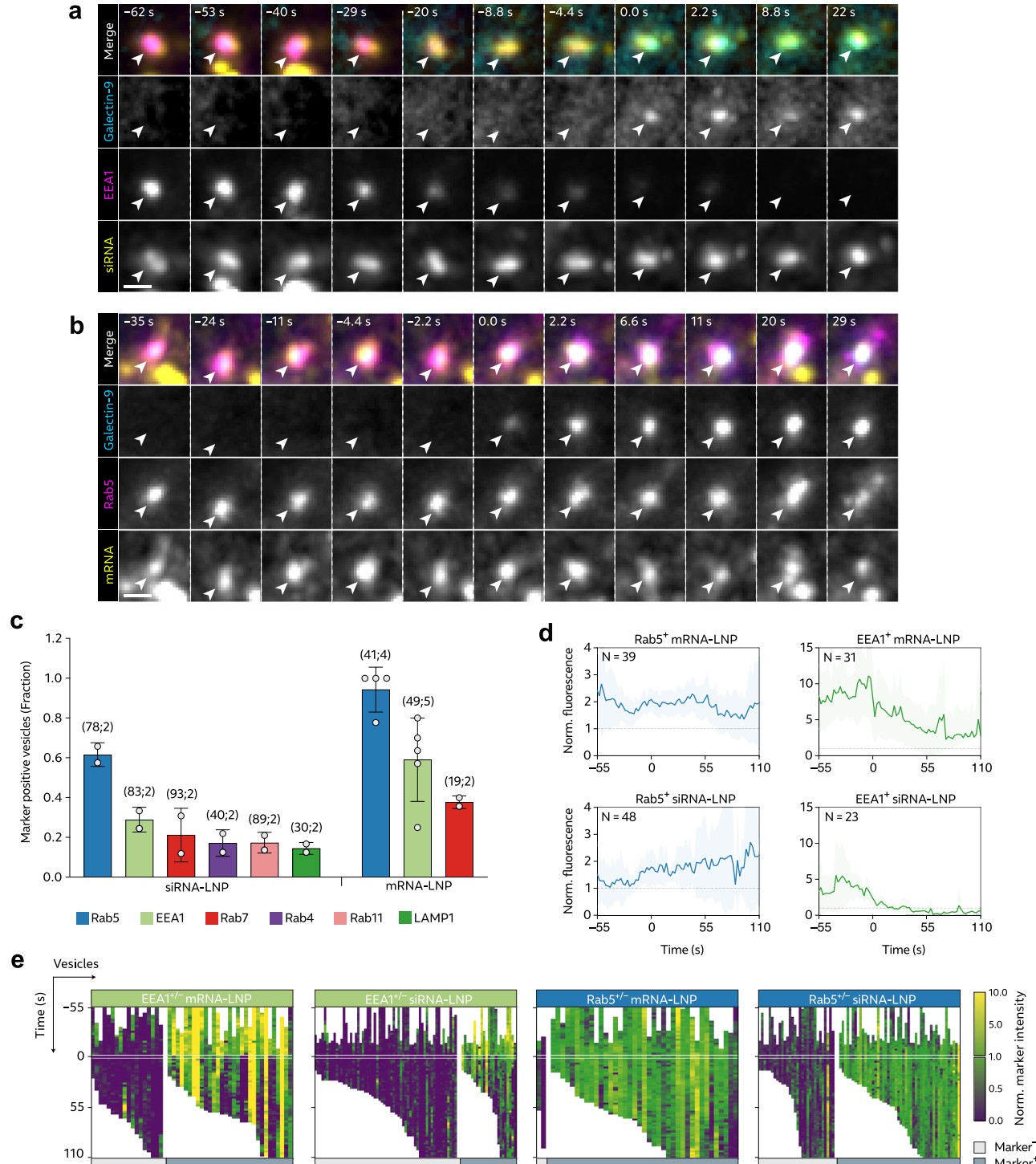

**Fig. 3 | LNPs damage Rab5⁺ EEA1⁺/⁻ early endosomes to promote cytosolic RNA release.** HeLa-Gal9-YFP cells expressing mScarlet- or mCherry-tagged markers of endosomal compartments were incubated with 50 nM AF647-siRNA-LNP or 0.75 μg mL⁻¹ Cy5-mRNA-LNP. Images were acquired with a high-speed widefield microscope, and events with de novo formation of galectin-9 foci were identified. **a** Galectin-9 is recruited to a siRNA-LNP containing endosome marked by EEA1, and **b** a mRNA-LNP containing endosome marked by Rab5. Images are representative of 109 EEA1⁺ and 74 Rab5⁺ events, from 2 and 4 independent experiments, for siRNA- and mRNA-LNP respectively. Scale bar is 2 μm. **c** LNP⁺ endosomes showing galectin-9 recruitment were tracked in 4D, measuring the endosomal marker fluorescence intensity. A dynamic intensity threshold (0.5 × 90th percentile intensity per cell) was applied to determine the fraction of vesicles with presence of the respective markers at a ~10 s time-interval centered around start of galectin-9 recruitment.

Bars show mean ± s.d. of 2–5 independent experiments (circles) per condition. Number of events (N_E) and experiments (N_X) shown in figure in parentheses as (N_E;N_X). Number of analyzed cells (order as **c**): 27, 23, 52, 14, 29, 15 and 25, 27, 13 cells for siRNA- and mRNA-LNP respectively and the indicated endosomal marker. **d** Rab5 or EEA1 fluorescence intensity measured on damaged LNP⁺ and Rab5⁺ or EEA1⁺ vesicles. Values were normalized to the 90th percentile marker intensity per cell (representing a typical marker-labeled object), and aligned in time so that $t = 0$ is the first time-point with galectin-9 recruitment. $N$ = vesicles, as indicated. Lines are mean; shades are 95% CI of mean. **e** Heatmaps showing all individual vesicles analyzed in (**c**) and (**d**). Endosomes are grouped as compartment marker⁻ or marker⁺ (left and right vertical subdivisions of maps, respectively), as described in (**c**). All traces were aligned in time (vertical) as described in (**d**). $N$ = vesicles as shown in (**c**).

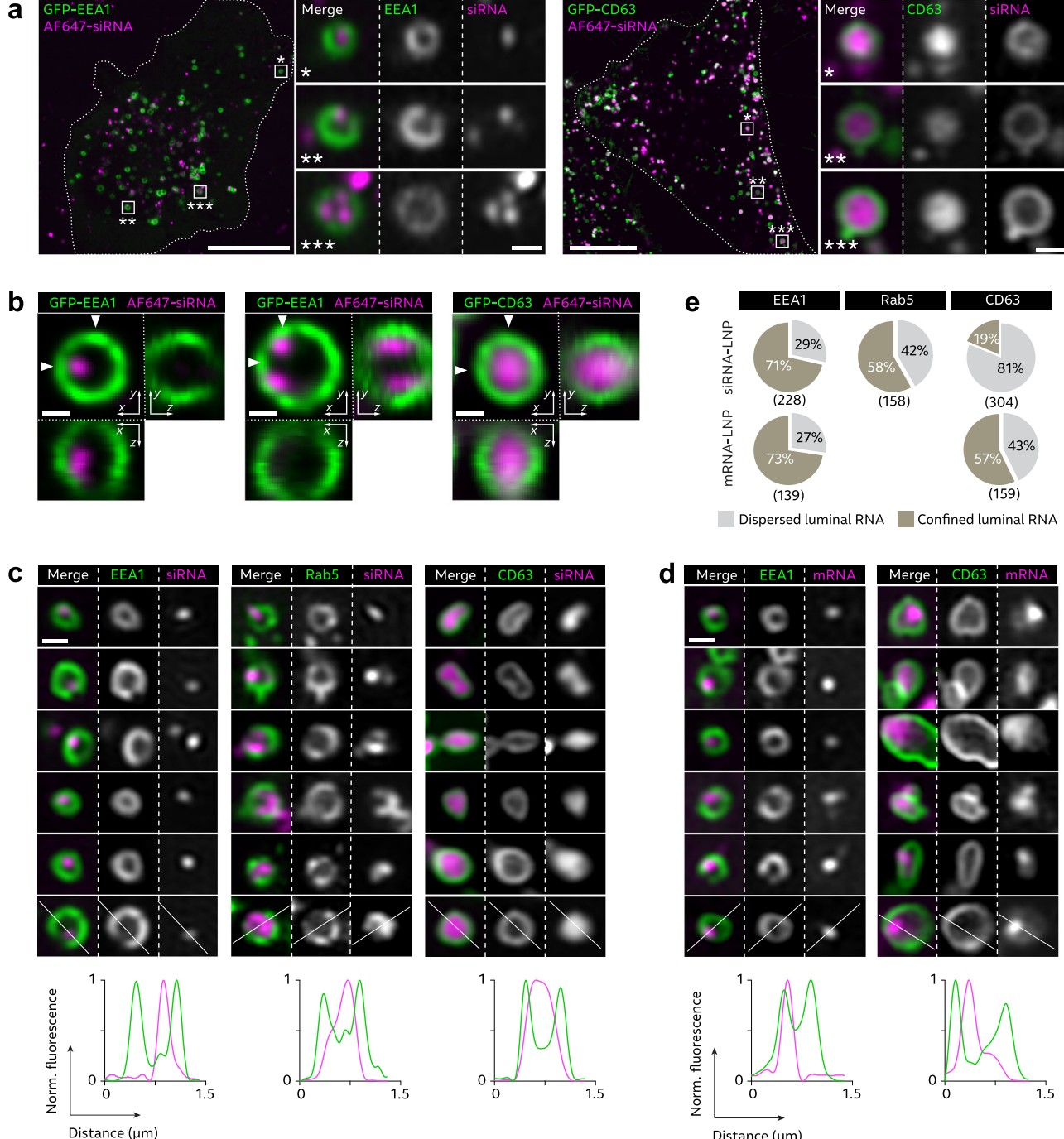

**Fig. 4 | RNA-LNPs disintegrate during endosomal maturation.** HeLa cells expressing GFP-EEA1, GFP-CD63 or GFP-Rab5 were incubated with 100 nM AF647-siRNA-LNP or 1.5 µg mL$^{-1}$ Cy5-mRNA-LNP, for approximately 15–75 min (EEA1), 45–120 min (CD63) or 30–90 min (Rab5). **a** Cells were imaged using live-cell Air-yscan confocal microscopy. Representative images showing well-defined siRNA-LNPs inside EEA1$^+$ endosomes, and CD63$^+$ endosomes with less confined siRNA fluorescence. Images are representative of 1 experiment. **b** z-stacks, with 21 slices and 100 nm step-size, were acquired using VT-iSIM. Arrows in the *xy* plane image indicate the positions where the *xz* and *yz* orthogonal planes intersect the *xy* plane, showing orientations and locations of the respective views. Images are repre-sentative of 2 independent experiments. **c** Representative image details showing differences in LNP appearance in endosomes marked by EEA1, Rab5 or CD63.

Images were acquired using live-cell Airyscan confocal microscopy. Curves show intensity line profiles, indicated by white lines in image details. Intensities were normalized to the maximum intensity value for each line profile. **d** Representative image details of EEA1$^+$ and CD63$^+$ endosomes with mRNA-LNP, highlighting a mixed LNP appearance with partly confined and dispersed intralumenal mRNA in CD63$^+$ endosomes. Images were acquired using live-cell Airyscan confocal microscopy. **a–d** Brightness and contrast were adjusted separately for all images. Scale bars, 10 µm; details, 500 nm. **e** Fraction of siRNA- and mRNA-LNP in EEA1$^+$, Rab5$^+$ or CD63$^+$ endosomes with primarily confined or dispersed appearance. $N$ = endosomes pooled from ~20–40 cells analyzed for each condition. **c–e** Images and data are from 4 (mRNA vs. CD63 and siRNA vs. CD63), 3 (siRNA vs. EEA1 and mRNA vs. EEA1) or 2 (siRNA vs. Rab5) independent experiments.

using the same composition as for MC3-LNPs (*i.e.*, BODIPY-MC3/DSPC/DMPE-PEG 50/38.5/10.1.5 mol%) in HeLa cells. Using RT-qPCR, we confirmed that the fully substituted BODIPY-MC3-LNPs with no conventional MC3 were functional, but ~5 times less effective than MC3-LNPs (Supplementary Fig. 9a). The internalization of BODIPY-MC3-LNPs during the first 4 h of incubation was approximately half that of MC3-LNP (Supplementary Fig. 9b), suggesting that the endosomal escape activity of BODIPY-MC3 was only around a factor two below that of MC3. As BODIPY was covalently coupled to one of the lipid tails with an ester bond, we wanted to confirm that BODIPY-MC3 stayed intact during our live-cell experiments. Using UHPLC-TOF-MS and fluorescence measurements we demonstrated low levels (4–7%) of free BODIPY acid after 1–4 h BODIPY-MC3-LNP incubation (Supplementary Fig. 9c). These results pave the way for the simultaneous study of RNA cargo and functional ionizable lipid.

Next, we set out to profile the fluorescence properties of the BODIPY-MC3 AF647-siRNA LNPs (Fig. 5a) and BODIPY-MC3 Cy5-mRNA LNPs (Supplementary Fig. 9d) both when intact and when disrupted by TX-100. Intact dual-labeled LNPs displayed significant Förster resonance energy transfer (FRET) to AF647/Cy5 when BODIPY was excited, which disappeared after LNP disintegration. Additionally, when intact, BODIPY-MC3 LNPs exhibited massive quenching of green BODIPY fluorescence (accompanied by red-shifted fluorescence emission) and slight quenching of AF647 fluorescence (similar to single-labeled AF647-siRNA-LNPs). These specific fluorescence properties allowed us to distinguish between intact and disintegrated LNPs. Intact LNPs were characterized by high FRET efficiency and low BODIPY signal, while disintegrated LNPs did not exhibit FRET but very high BODIPY fluorescence. In live HeLa cells, intact LNPs displaying low BODIPY fluorescence and efficient FRET were observed primarily in the periphery of cells (Fig. 5b). On the contrary, FRET⁺ structures were scarce in perinuclear regions, while instead structures with high BODIPY intensity were typically present. To obtain quantifiable measures, we determined the distance for each individual free BODIPY-MC3⁺ spot, red-shifted BODIPY-MC3⁺ spot and FRET⁺ spot from the nucleus, and normalized to the outermost spot within each cell. This indicates that disintegrated LNPs were more frequently located near the nucleus, likely reflecting their presence in later endolysosomes, further supporting the conclusion that both siRNA- and mRNA-LNPs disintegrate during endosomal maturation. Moreover, we observed that larger and more intense redshifted (dense) BODIPY-MC3⁺ structures were more frequent in the perinuclear regions for mRNA-LNPs compared to siRNA-LNPs (Fig. 5b). The fluorescence emission spectrum of BODIPY is redshifted from green (515 nm emission maximum) to red (620 nm emission maximum) wavelengths at high concentrations[37], suggesting that endosomes containing denser, more aggregated ionizable lipid are more common in late endosomes for mRNA-LNPs, consistent with the observed mRNA-cargo aggregates also seen (Fig. 4d).

Intriguingly, we observed structures in the perinuclear region that appeared to be either exclusively BODIPY⁺ or AF647/Cy5⁺ (Fig. 6a). Building on our previous finding that endosomes containing free BODIPY-MC3 are more frequently localized in perinuclear regions compared to those containing dense BODIPY-MC3, we examined the colocalization between free BODIPY-MC3⁺ or dense BODIPY-MC3⁺ and RNA⁺ endosomes. Endosomes with dense BODIPY-MC3 signal more frequently contained RNA compared to endosomes containing free BODIPY-MC3 (Fig. 6b), indicating that RNA⁻ vesicles, or endosomes with highly diluted RNA, are more prevalent in late endolysosomes. However, the reported level of colocalization between either free BODIPY-MC3 or dense BODIPY-MC3 and RNA is likely understated due to sensitivity limitations of the microscopy assay and channel misalignment caused by time delays during filter changes. As the great majority of intact single LNPs contained both BODIPY-MC3 and AF647-siRNA (Supplementary Fig. 9e), these findings provide evidence for segregation of BODIPY-MC3 and RNA payload during endosomal

trafficking. Taken together, this suggests that the significant fraction of endosomal damages appearing in vesicles without appreciable RNA cargo are caused by presence of ionizable lipid alone, after segregation of the lipid and RNA constituents.

In a subset of endosomal structures, BODIPY-MC3 appeared to be distributed within the vesicle membrane while siRNA was either homogenously dispersed in the endosomal lumen or tethered to the endosomal membrane as a confined particulate focus, demonstrating sub-endosomal separation of BODIPY-MC3 and the RNA payload (Fig. 6c and Supplementary Fig. 9f). Furthermore, regions with red-shifted BODIPY fluorescence were observed in endosomal membranes enriched in BODIPY-MC3, indicating local accumulation of BODIPY-MC3. Interestingly, the BODIPY-MC3 rich domains were regularly in close proximity to membrane-tethered siRNA-LNP remnants. In Rab5⁺ vesicles, the primary compartment of LNP-mediated membrane disruption, BODIPY-MC3 was frequently associated with the endosomal membrane (Fig. 6d). This indicates that integration of ionizable lipid into the membrane of early endosomes promotes membrane destabilization and disruption with potential RNA release.

## Membrane-tethering of LNPs is associated with localized membrane destabilization

Next, we wanted to assess if polarized LNP-RNA localization—with LNPs appearing close or tethered to the endosomal membrane—was associated with localized membrane destabilization as marked by recruitment of CHMP2A or galectin-9. In these settings, cell fixation was necessary to acquire reliable super-resolution microscopy images. First, we confirmed that the appearance of intact LNPs (with a polarized sub-endosomal localization) and disintegrated LNPs (with homogenously dispersed luminal siRNA signal) were maintained after PFA fixation, in early and late endosomes, respectively (Supplementary Fig. 10). The identity of compartments to where CHMP2A is recruited during LNP treatment is not known. However, it has been suggested that ESCRT components are recruited to sites of endolysosomal membrane damage before accumulation of galectins[28]. As a consequence, we decided to examine damage response in both EEA1⁺ and Rab5⁺ early endosomes. Interestingly, both CHMP2A and galectin-9 mainly localized to endosomal membrane domains in direct proximity to the polarized siRNA-LNP (Fig. 7a, b). This was observed both in endosomes containing a single LNP associated with a single site of membrane damage and in endosomes containing two LNPs with two corresponding damage sites. In a subset of endosomes with multiple LNPs, localized galectin recruitment revealed which LNP that likely initiated membrane damage (Fig. 7b). We devised a custom quantitative analysis to compare the sub-endosomal localization of damage markers in relation to LNP-siRNA. Briefly, for CHMP2A or galectin-9 and LNP-siRNA foci respectively, vectors were defined using the center of endosomes as initial points and the local intensity maxima of the foci as terminal points (Fig. 7c). To describe the proximity of the local maxima (damage marker versus siRNA), we calculated the angular difference between the vectors. If their localization were random, the angular difference would be evenly distributed with an average 90° if comparing one paired local maximum per endosome, or 45° if comparing two paired maxima. With both CHMP2A and galectin-9, the median angular difference to the siRNA was small (single LNP, CHMP2A/EEA1: 30°; CHMP2A/Rab5: 17°; galectin-9/Rab5: 26°), and the great majority of measurements were considerably less than 90° (single maximum) or 45° (two maxima) (Fig. 7d). Comparing the magnitude of the vectors showed that both CHMP2A and galectin-9 maxima were typically located further away from the center of the endosome than the siRNA (median difference, CHMP2A/EEA1: 69 nm; CHMP2A/Rab5: 65 nm; galectin-9/Rab5: 83 nm) (Fig. 7e), indicating a more exterior interaction between the damage markers and endosomal membrane compared to the luminally tethered siRNA-LNP.

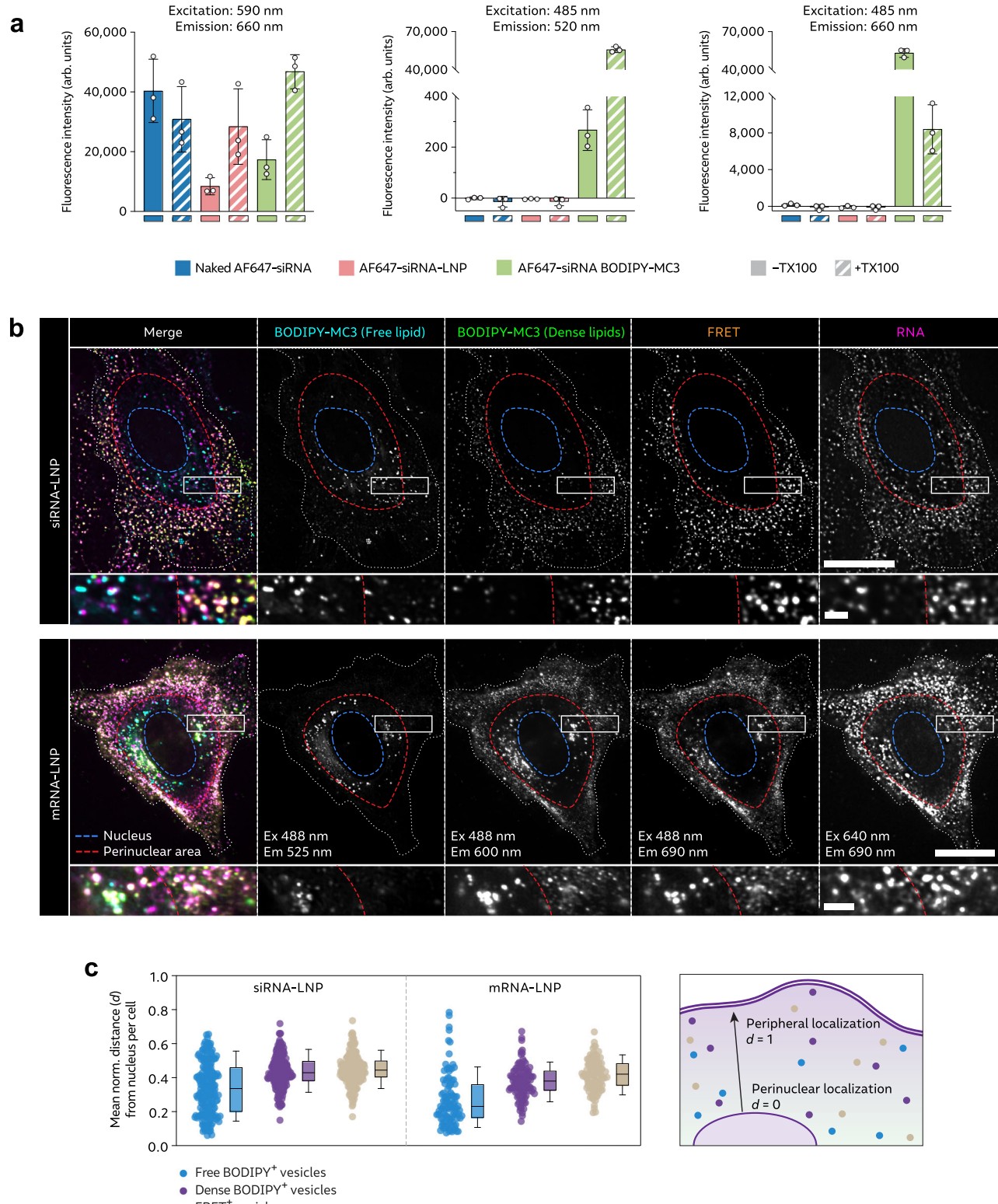

**Fig. 5 | Fluorescence profiling of BODIPY-MC3 containing LNPs reveals different spatial distribution of intact and disintegrated LNPs. a** Fluorescence intensity of naked AF647-siRNA, AF647-siRNA-LNPs and AF647-siRNA BODIPY-MC3 LNPs (all 50 nM RNA concentration) at the indicated excitation/emission wavelengths, measured with a spectrophotometer. Triton-X-100 (TX100, 1% final concentration) was added to disrupt LNPs. Bars show mean ± s.d. of 3 independent experiments (circles). **b, c** HeLa wildtype cells were incubated with 100 nM AF647-siRNA BODIPY-MC3 LNPs or 1.5 µg mL⁻¹ Cy5-mRNA BODIPY-MC3 LNPs for 1–4 h and imaged using VT-iSIM. **b** Representative images of cells illustrating spatial distribution differences across channels. White outlines indicate cell borders, red outlines show perinuclear regions, and blue outlines indicate nuclei. Brightness and contrast were adjusted separately for all images. Scale bars, 20 µm; details, 2 µm. **c** For all free BODIPY-MC3⁺, redshifted dense BODIPY-MC3⁺, and FRET⁺ foci, the distance (*d*) from the nucleus was quantified and normalized to the inner- and outermost foci in each cell. The mean distance between foci and nucleus per cell was calculated for each channel. Circles are cells. *N* = 198 and 104 cells for siRNA- and mRNA-LNP, respectively. Boxes are median ± i.q.r.; whiskers are 10–90 percentiles. Data and images are from 2 independent experiments.

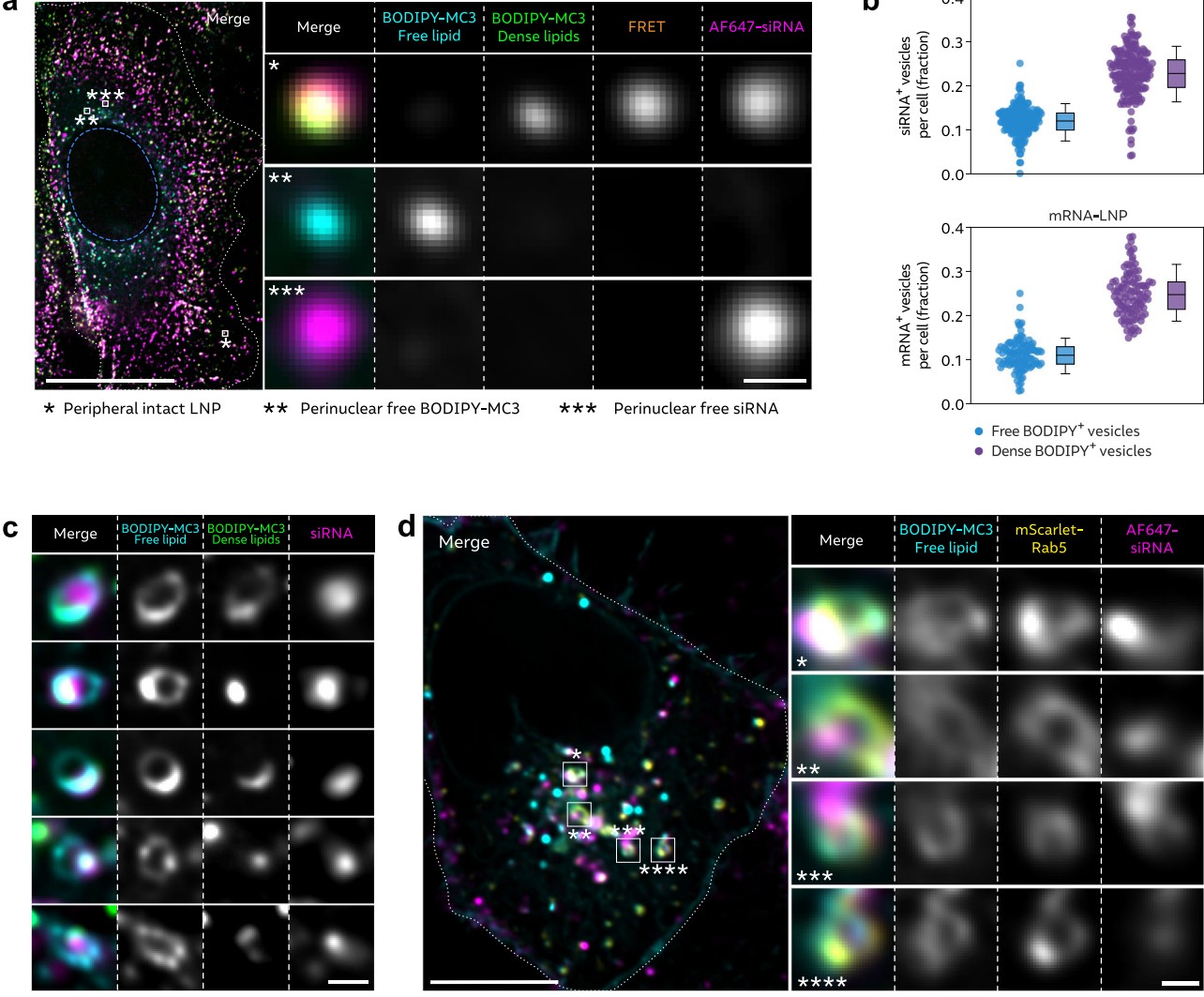

**Fig. 6 | BODIPY-MC3 and RNA are partially separated during endosomal sorting.** HeLa wildtype cells (**a**–**c**) or HeLa cells expressing mScarlet-Rab5 (**d**) were treated with 100 nM AF647-siRNA BODIPY-MC3 LNPs or 1.5 μg mL⁻¹ Cy5-mRNA BODIPY-MC3 LNPs for 1–4 h, and then imaged using a VT-iSIM. **a** Representative images showing BODIPY-MC3⁺RNA⁺, BODIPY-MC3⁺RNA⁻, BODIPY-MC3⁻RNA⁺ endosomes. White outlines indicate cell borders; blue outlines indicate nucleus. Scale bars, 30 μm; details, 500 nm. **b** The fraction free BODIPY-MC3⁺ or redshifted dense BODIPY-MC3⁺ colocalizing with RNA⁺ foci was quantified, with colocalization defined as >25% overlap. The fraction of RNA⁺ endosomes was calculated for each cells. Circles are cells. *N* = 198 and 104 cells for siRNA- and mRNA-LNP, respectively. Boxes are median ± i.q.r.; whiskers are 10–90 percentiles. Data and images are from 2 independent experiments. **c** Representative examples of endosomes with BODIPY-MC3 association with endosomal membranes. Brightness and contrast were adjusted separately for all images. Scale bar is 1 μm. **d** Representative images illustrating BODIPY-MC3 integration into Rab5⁺ endosomal membranes. Scale bars, 10 μm; details, 1 μm. Images are representative of 2 independent experiments.

As shown above, BODIPY-MC3 accumulate in endosomal membrane nanodomains adjacent to siRNA-LNP remnants (Fig. 6c) and tethering of siRNA-LNPs to the luminal endosomal membrane is associated with localized recruitment of membrane damage response systems. In addition, we also observed that galectin-9 was frequently recruited to endosomal membrane nanodomains where high concentrations of BODIPY-MC3 and tethered siRNA localized (Fig. 7f) in cells treated with BODIPY-MC3 AF647-siRNA LNPs. Collectively, we provide evidence that local enrichment of ionizable lipid close to the membrane-tethered RNA is likely instrumental in promoting membrane destabilization and damage.

## Discussion

Here, we identified multiple distinct steps of substantial inefficiency for the cytosolic delivery of LNP-RNA payload, that have previously not been characterized. Using high-speed live-cell microscopy, we found that only a small fraction of the RNA payload in endosomes disrupted during LNP treatment was released to the cytosol. Intriguingly, a significant fraction of damaged endosomes did not contain appreciable RNA payload. Super-resolution live-cell microscopy visualized gradual disintegration of RNA-LNPs during endosomal trafficking and partial segregation of the ionizable lipid and RNA payload into separate endosomal compartments. In addition, during this segregation process we observe localized membrane damage and ionizable lipid enrichment in membranes of early endosomes, near membrane-tethered intralumenal RNA-LNPs.

Galectins have recently gained widespread use as a tool to investigate the intracellular delivery of RNA therapeutics, both with LNPs[24–26,38] and other strategies[39]. Our findings show that broadly evaluating galectin-9 response as a proxy of successful cytosolic delivery of RNA-LNP payload is an oversimplification, as the release efficiency and hit rate (cargo load) of damaged vesicles is not taken into account. Both parameters are likely to vary between different RNA-LNP formulations, and will constitute additional complication for

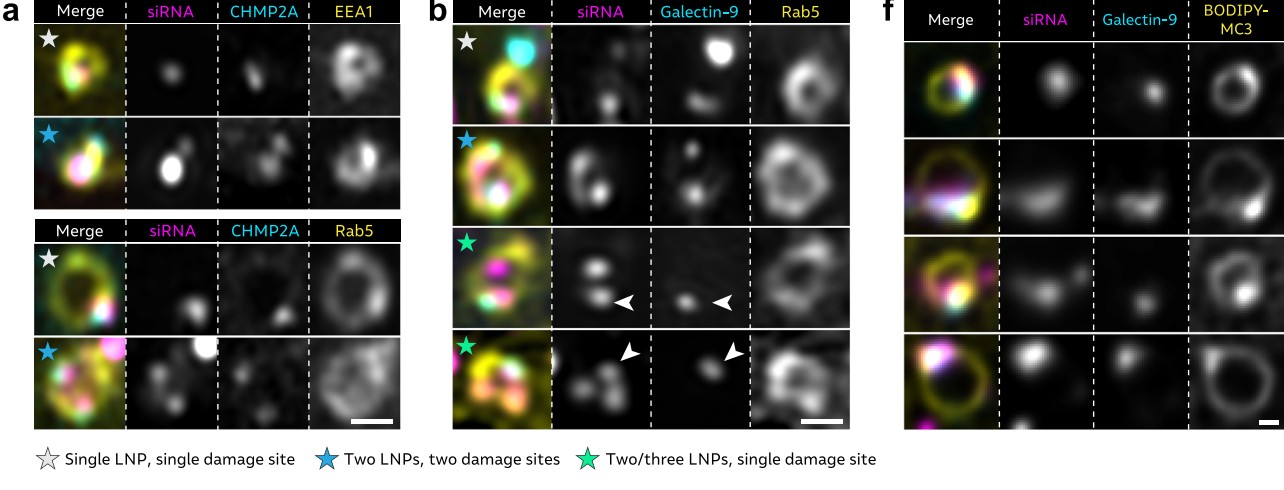

☆ Single LNP, single damage site    ★ Two LNPs, two damage sites    ★ Two/three LNPs, single damage site

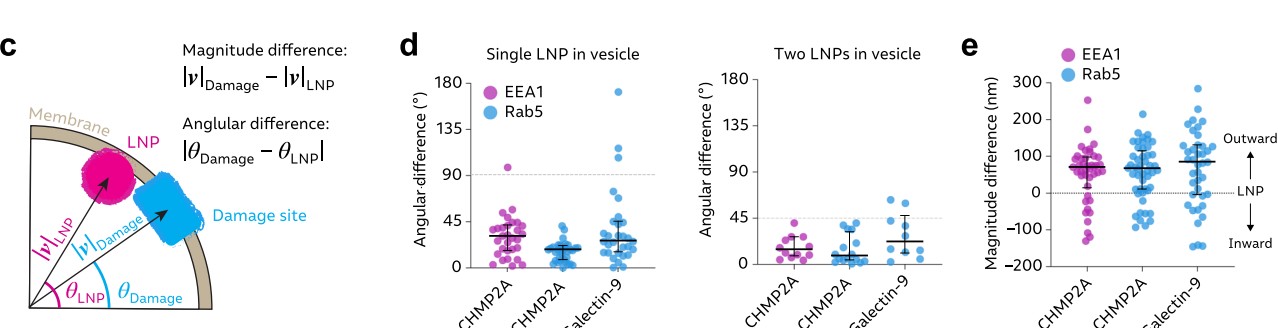

**Fig. 7 | Membrane-tethered intralumenal siRNA-LNP colocalize with recruited membrane damage sensors.** HeLa cells expressing GFP-CHMP2A and mScarlet-EEA1 or mScarlet-Rab5, or YFP-galectin-9 and mScarlet-Rab5 were incubated with 100 nM AF647¼-siRNA-LNPs for 90 min. Cells were fixed with PFA and imaged using Airyscan confocal microscopy. **a**, **b** Representative image details showing the spatial relationship of CHMP2A or galectin-9 recruitment to single or multiple membrane-adjacent intralumenal AF647-siRNA-LNP (indicated by color-coded stars). Brightness and contrast were adjusted individually for all images. Scale bars are 500 nm. **c**–**e** EEA1⁺ or Rab5⁺ endosomes with membrane-adjacent intralumenal siRNA-LNP and membrane-associated CHMP2A or galectin-9 were identified, and a vector from the center of the endosome to the respective focus was computed. The vector magnitude difference and angular difference between the LNP and galectin-9 or CHMP2A vectors were calculated, in endosomes with one or two pairs of siRNA-LNP and CHMP2A or galectin-9 maxima. **d** Calculated angular difference between LNP and galectin-9 or CHMP2A vectors, in endosomes with one or two

paired siRNA-LNP and CHMP2A or galectin-9 maxima. Circles are paired vectors (pooled), N = 30, 33 or 30 single paired maxima and 12, 16 or 10 dual paired maxima, for CHMP2A vs. EEA1, CHMP2A vs. Rab5 and galectin-9 vs. Rab5, respectively. Bars are median ± i.q.r. **e** Calculated vector magnitude difference between CHMP2A or galectin-9 and siRNA-LNP vectors. Values > 0 indicate greater distance from center of endosome to CHMP2A or galectin-9 focus than to siRNA-LNP focus. Bars are median ± i.q.r. Circles are paired vectors (pooled), N = 42, 49 or 40 paired maxima for CHMP2A vs. EEA1, CHMP2A vs. Rab5 and galectin-9 vs. Rab5, respectively. **a**–**e** All data and images are from 2 (CHMP2A vs. EEA1) or 3 (CHMP2A vs. Rab5 and galectin-9 vs. Rab5) independent experiments, with 15–20 cells analyzed for each condition. **f** HeLa cells expressing mCherry-galectin-9 were treated with 100 nM AF647-siRNA BODIPY-MC3 LNPs for 1.5–4 h and imaged continuously using VT-iSIM. BODIPY-MC3 (free lipid) was acquired at 488/525 nm excitation/emission. Images are representative of 2 independent experiments. Brightness and contrast were adjusted separately for all images. Scale bar is 500 nm.

efforts to deliver multi-component RNA formulations for, e.g., gene editing.

Several factors might influence the observed damages to RNA depleted endosomes, and the apparent difference in hit rate between siRNA- and mRNA-LNPs. Even though there are limitations to the sensitivity of our microscopy assay, in detecting single or very few siRNA or mRNA molecules, we were indeed able to detect intact single LNPs with high confidence. Together with the observed segregation of ionizable lipid and RNA into separate endosomal compartments, this strongly suggests that LNP-induced disruption of endosomes with very small or no RNA payload is a genuine phenomenon, and possibly mechanistically distinct from damages adjacent to semi-intact LNPs. It is known that alterations in LNP lipid composition and the binding to siRNA or mRNA payload itself influence LNP stability and rate of LNP disintegration[40]. It is conceivable that this could translate into variations in compartmental segregation of ionizable lipid and cargo, possibly contributing to hit rate variability between LNP formulations. The alternative hypothesis that a large fraction of vesicles exhibiting

galectin recruitment has already released all RNA cargo is not deemed probable given the strict correlation of galectin damages and cytosolic siRNA release (99% detection sensitivity) seen with cationic lipids[33] and the lack of release from ESCRT positive perturbations seen here. Indeed, it is also possible that part of the cargo is exocytosed while a fraction of the membrane destabilizing lipid remains inside the cell[36].

The exact nature of the LNP-induced membrane perturbations allowing for intralumenal RNA payload to escape is still not clear, and several biophysical models can be considered. Protonation of the ionizable lipid and its interaction with the endosomal membrane bilayer, transitioning into hexagonal lipid phase, is widely considered a key step preceding cargo release[22,35]. Galectins, used here to successfully capture sudden but seemingly transient RNA escape from a subset of damaged vesicles, are known to respond to disruptions in endosomal membrane permitting release of macromolecular fluid-phase cargo like dextran (with a molecular weight comparable to siRNA) when damages are induced by for example small molecule drugs[32,41]. Since galectins bind to β-galactoside-containing glycans on

proteins present on the luminal side of the lipid bilayer[31,42], membrane integrity disruptions marked by galectin recruitment are believed to be more extensive (larger membrane defects) than perturbations sensed by primarily the ESCRT machinery[27]. It is proposed that mixing of ionizable lipid and RNA with the endosomal membrane could serve as a conduit for RNA transitioning to the cytosol, without the need of large membrane defects[29]. While ESCRT components conceivably could respond to such membrane perturbation, it is unclear if it would be sufficient to trigger galectin recruitment as observed during RNA release captured here. Given our observations, we propose that mobile intralumenal RNA, resulting from partial LNP disintegration, is released through disruptions in the endosomal membrane following its destabilization by enrichment of ionizable lipid into the lipid bilayer. In that case, the transient and incomplete RNA release could be attributed to RNA molecules remaining trapped in the partially disintegrated LNP remnant. We also show that disintegrated mRNA-LNPs appeared more membrane-tethered, with less free dispersed intralumenal RNA compared to siRNA-LNPs, possibly explaining the smaller fraction of endosomes exhibiting rapid release of mRNA-LNPs observed here. Similar differences in disintegration were reported in a recent study using early endosomal membrane mimics, where siRNA was fully released upon LNP fusion with an anionic supported lipid bilayer, while mRNA exhibited partial release, with a fraction of mRNA remaining membrane associated[43]. This aligns well with our imaging data, demonstrating that siRNA-LNP disintegrate more completely, while mRNA-LNPs remain partially membrane-associated, limiting RNA dispersion. Alternatively, the incomplete RNA release could be explained by membrane damages being transient and possibly limited by self-sealing of the membrane or recruitment of cytosolic damage responders acting as a seal and repair system. Until more is known about the precise nature of membrane disruption, mechanisms by which RNA could escape the endosome that are not recognized by galectins, with contribution to the biological effect, cannot be completely ruled out.

We demonstrate that a subset of membrane perturbations induced by RNA-LNPs are detected by the ESCRT-III machinery but not galectins. Endosomal disruptions of this kind did not promote cytosolic release of siRNA, possibly because they are too small to permit release of RNA payload or are quickly sealed by the ESCRT membrane repair system before disruption accelerates and release is possible. Knockdown of key ESCRT components made cells more prone to experience galectin-9[+] membrane damages, possibly because smaller membrane perturbations that are normally sealed by ESCRT machinery accumulated and escalated to larger disruptions when unrepaired. This progression highlights the role of galectin-9 as a marker for more extensive membrane disruptions. However, we do not yet know the RNA release efficiency of the additional galectin-9[+] damage events observed during ESCRT depletion.

For the first time, we captured the dynamics of mRNA-LNP-mediated endosomal damage and release of mRNA payload in living cells. Earlier studies have used single-molecule localization microscopy of fixed cells to investigate the egress of mRNA from early endosomes with nanometer spatial resolution[44], but without temporal or quantitative information such characterization is hard to interpret or use for evaluating the delivery efficiency. High-speed imaging of LNP-mediated membrane damage and endosomal escape, with spatial resolution beyond the diffraction limit in live cells, remains a major technical challenge where further development is needed. However, here we visualize RNA-LNP disintegration with subendosomal resolution in living cells. LNP interaction with membranes as well as particle disintegration has previously been studied using endosomal model membranes[35] with LNP-membrane binding demonstrated already at pH 6.6, consistent with the high degree of membrane tethering of LNPs in EEA1[+] early endosomes shown here.

Lastly, we recognize that other ionizable lipids, such as ALC-0315 (used in the Pfizer/BioNTech covid-19 vaccine formulation), or SM-102 (used by Moderna), are more specifically optimized for mRNA delivery. Therefore, it will be interesting to explore how these, and other lipids, might impact LNP disintegration, intracellular trafficking, and RNA release dynamics using the tools presented here.

In conclusion, we hope that future efforts will further increase the mechanistic understanding of the fundamental barriers limiting the intracellular delivery of LNP payload uncovered in this work, ultimately improving the design rationale and efficacy of current LNP therapeutics to successfully reach new extrahepatic targets.

## Methods

### Cell culture and reagents

HeLa cells were purchased from the American Type Culture Collection and were confirmed to be free from Mycoplasma. HeLa cells with stable expression of YFP-tagged galectin-9 (HeLa-Gal9-YFP)[32] and destabilized eGFP (HeLa-d1-eGFP)[33] were described previously. Cells were cultivated in Dulbecco's Modified Eagle Medium (DMEM) (Cytiva HyClone Laboratories, South Logan, UT, USA) supplemented with 10% fetal bovine serum (FBS, Gibco), 100 U mL$^{-1}$ penicillin and 100 mg mL$^{-1}$ streptomycin (Gibco) and 2 mM glutamine (Gibco) at 37 °C and 5% $CO_2$. Imaging medium was used for live-cell microscopy experiments, consisting of FluoroBrite DMEM (Gibco), supplemented with 10% FBS, 2 mM glutamine and 2 mM HEPES (Thermo Fisher Scientific, Waltham, MA, USA). Cell culture medium was supplemented with 1.5 μg mL$^{-1}$ recombinant ApoE (Sigma), prepared according to manufacturer's protocol, for all LNP experiments. For microscopy experiments, 4–5 × 10$^4$ cells were plated per well in 8-well Lab-Tek II Chamber Slides (Nunc, Rochester, NY, USA) 1 day before experiments. A Neon Transfection System (Thermo Fisher Scientific) was used for plasmid and siRNA transfection. A total of 5 × 10$^5$ cells were used with the 100-μL tips and cell-type specific protocol provided by the manufacturer. Transfected cells were used in experiments between 24 and 48 h after transfection.

### Plasmids, siRNA, mRNA and primers

CHMP2A_GFP_N_term was a gift from Daniel Gerlich (Addgene plasmid # 31805). Lamp1-mScarlet-I (Addgene plasmid # 98827) and mScarletI-Rab7 (Addgene plasmid # 112955) were gifts from Dorus Gadella. mCherry-Rab4a-7 (Addgene plasmid # 55125) and mCherry-Rab11a-7 (Addgene plasmid # 55124) were gifts from Michael Davidson. mScarlet-EEA1 (Addgene plasmid # 169067) and GFP-Rab5B (Addgene plasmid # 61802) were gifts from Gia Voeltz. GFP-EEA1 wt was a gift from Silvia Corvera (Addgene plasmid # 42307). PB-CAG-mRuby3-Gal8-P2A-Zeo was a gift from Jordan Green (Addgene plasmid # 150815). GFP-CD63 was a gift from Judy Lieberman. mScarlet-Rab5 was subcloned from mCherry-Rab5a-7 (gift from Michael Davidson, Addgene plasmid # 55126) into pmScarlet-i_C1 (gift from Dorus Gadella, Addgene plasmid # 85044). mCherry-Galectin-9 was produced by cloning LGALS9 transcript 1 into pEZ-M55 (GeneCopoeia, Rockville, MD, USA).

The following siRNAs were used: AF647-siGFP sense: 5′- GGC UAC GUC CAG GAG CGC Atst-AF647 -3′, AF647-siGFP antisense: 5′- UGC GCU CCU GGA CGU AGC Ctst -3′; Cy5-siGFP sense: 5′- GGC UAC GUC CAG GAG CGC Atst-Cy5 -3′, Cy5-siGFP antisense: 5′- UGC GCU CCU GGA CGU AGC Ctst -3′; unlabeled siGFP sense: 5′- GGC UAC GUC CAG GAG CGC Ats t-3′, unlabeled siGFP antisense: 5′- UGC GCU CCU GGA CGU AGC Ctst -3′. Lowercase denotes deoxynucleotides and 's' indicates phosphorothioate linkage. All siRNAs targeting eGFP were custom synthesized by Integrated DNA Technologies Inc, Coralville, IA, USA. CHMP6 (Hs_CHMP6_1) target sequence: 5′- CTG AGC GCA ATC ACT CAG GAA -3′ (QIAGEN Sciences LLC, Germantown, MD, USA, cat. no. SI04218032); Tsg101 (Hs_TSG101_6) target sequence: 5′- CAG TTT ATC ATT CAA GTG TAA -3′ (QIAGEN, cat. no. SI02655184); Alix

(Hs_PCD6IP_5) target sequence: 5′- AAG AGC TGT GTG TTG TTC AAT -3′ (QIAGEN, cat. no. SI02655345); Negative Control siRNA, proprietary target sequence (QIAGEN, cat. no. 1027281).

The following PCR primers were used: eGFP forward: 5′- ACG TAA ACG GCC ACA AGT TC -3′, eGFP reverse: 5′- AAG TCG TGC TGC TTC ATG TG -3′ (Sigma); GAPDH forward: 5′- CTG GGC TAC ACT GAG CAC C -3′, GAPDH reverse: 5′- AAG TGG TCG TTG AGG GCA ATG -3′ (Invitrogen, Thermo Fisher Scientific); ALIX forward: 5′- ATG GCG ACA TTC ATC TCG GTG -3′, ALIX reverse: 5′- CGC TTG GGT AAG TCT GCT GG -3′ (IDT); TSG101 forward: 5′- ATG GCT ACT GGA CAC ATA CCC -3′, TSG101 reverse: 5′- GCG GAT AGG ATG CCG AAA TAG -3′ (IDT); CHMP6 forward: 5′- TTG AGT TCA CCC AGA TCG AAA TG -3′, CHMP6 reverse: 5′- TGG CAG CTC TAT TTG TTC CTG -3′ (IDT).

CleanCap, 5-methoxyuridine (5moU) modified mRNA was purchased from TriLink BioTechnologies (San Diego, CA, USA), encoding eGFP (cat. no. L-7201) or Cy5-labeled eGFP (cat. no. L-7701). Cy5-labeled mRNA contained 25% of the Cy5-uridine substituting 5moU, statistically distributed throughout the sequence.

### Lipid nanoparticle formulation

Cationic ionizable lipid (6Z,9Z,28Z,31Z)-Heptatriaconta-6,9,28,31-tetraen-19-yl 4-(dimethylamino)butanoate (Dlin-MC3-DMA) was synthesized at AstraZeneca according to literature[19]. BODIPY-MC3 was synthesized at AstraZeneca as described in Supplementary Note 1. Helper lipids were purchased from commercial vendors, Cholesterol (Sigma-Aldrich), 1,2-distearoyl-sn-glycero-3-phosphocholine (DSPC; CordenPharma), 1,2-dimyristoyl-sn-glycero-3-phosphoethanolamine-N [methoxy (polyethylene glycol)-2000] (DMPE-PEG2000; NOF Corporation).

Lipid nanoparticle formulations were prepared using a NanoAssemblr Benchtop microfluidic mixer (Precision Nanosystems) by mixing at a 3:1 volumetric ratio (aqueous:ethanol) and 12 mL min$^{-1}$ total flow rate. Lipid solutions were made in pure ethanol at a molar ratio of (50:10:38.5:1.5 for ionizable lipid:DSPC:cholesterol:DMPE-PEG2000) at a concentration of 12.5 mM. mRNA solutions were prepared using nuclease free reagents with final buffer concentration of 50 mM citrate pH 3. For mRNA containing formulations, a w:w ratio of excipient to cargo of 10:1 was used, for siRNA containing formulations a N/P ratio of 3 was used.

Immediately after microfluidic mixing, LNPs were transferred to 10 kDa MWCO Slide-A-Lyzer G2 Dialysis Cassettes (ThermoFisher Scientific) and dialyzed overnight against a 1000-fold volume of 20 mM Tris pH 7.5. After dialysis, formulations were removed from the cassette with a syringe and needle and then sterile filtered through a 0.2 μm filter (hydrophilic PTFE, Millex LG). Filter sterilized sucrose solution was added to a final concentration of 8% (w/v) in 20 mM Tris pH 7.5 and LNPs were concentrated by centrifuging at 3000 × g in Amicon Ultra centrifugal filter units with a 100 kDa MWCO (Millipore Sigma) to a concentration of 0.3–0.4 mg mL$^{-1}$ mRNA or approximately 20 μM siRNA. LNPs were characterized by dynamic light scattering using a Zetasizer Nano ZSP (Malvern Analytical Inc.) mRNA concentration and encapsulation efficiency measurement was performed using Quant-iT RiboGreen reagent (Invitrogen). Total concentration of siRNA was measured by analyzing fluorescence intensity of the AlexaFluor647 or Cy5 label and a corresponding standard curve.

### BODIPY-MC3 stability (LC-MS)

HeLa cells were seeded in 24-well plates, $1 \times 10^5$ cells per well. The following day, cells were treated with 100 nM AF647-siRNA LNPs or AF647-siRNA BODIPY-MC3 LNPs. Control cells were treated with imaging medium supplemented with rApoE. Three technical replicates were used. Cells were harvested after 1, 2 or 4 h, when they were washed once with PBS, detached through trypsinization, diluted in imaging medium, transferred to cryovials and centrifuged at 400 × g for 5 min. The supernatant was removed and pelleted cells were snap

frozen by submerging the cryovial in liquid nitrogen for 60 s. Samples were then placed at –80 °C overnight and shipped to the analysis facility (AstraZeneca, Gothenburg, Sweden).

The cellular uptake of MC3 and BODIPY-MC3 was measured using UHPLC-TOF-MS. Briefly, one-phase extraction was performed by adding 500 μL cold isopropanol containing 0.1% formic acid and internal standard (0.5 μg mL$^{-1}$ DLin-DMA) to each cell sample. The suspension was incubated for 30 min in a mechanic shaker at 400 rpm followed by 15 min of centrifugation at 12,390 × g and 10 °C. 300 μL of the liquid phase was carefully transferred to an LC vial. The DLin-MC3-DMA samples were diluted in 0.1% formic acid in ethanol whereas MC3-BODIPY lipid extracts were diluted 1:1 in same diluent prior to analysis. The lipid quantitation was performed on a binary Agilent 1290 Infinity II system coupled to a Bruker Maxis II ETD Time-Of-Flight mass spectrometer operated in positive scanning mode. The analytical column was an AtlantisTM Premier BEH C18 AX (2.1 × 100 mm, 1.7 μm) maintained at 60 °C. Mobile phase A was 0.05% formic acid in 50:49:1 acetonitrile/water/0.5 M ammonium formate (v/v/v), and mobile phase B was 0.05% formic acid in 90:9:1 isopropanol/acetonitrile/0.5 M ammonium formate (v/v/v). The eluting gradient applied was: Time 0–0.5 min, 10% B; 0.5–3.0 min, 10–50% B; 3.0–9.0 min, 50–80% B; 9.0–9.5 min, 80% B; 9.5–10.5 min, 80–99% B; 10.5–11.5, 99% B; 11.5–12.0 min, 99–10% B; 12.0–17.0 min, 10% B. The flow rate and autosampler temperatures were kept at 0.4 mL min$^{-1}$ and 10 °C respectively. The sample lipid content was determined against external calibration curve based on extracted ion chromatogram, $[M + H]^+$, with internal standard normalization.

The level of free BODIPY acid (hydrolyzed MC3-BODIPY) was assessed using a Waters UPLC H-class chromatographic system with a fluorescence detector attached to it. The excitation and emission wavelengths for the fluorescence detector were set to 500 and 510 nm respectively. The peaks due to BODIPY acid, following hydrolysis of MC3-BODIPY, and the parent lipid were integrated.

### LNP internalization and galectin-9 response

HeLa-Gal9-YFP cells were incubated with 4, 20, 50 or 100 nM AF647-siRNA-LNP, or 0.06, 0.3, 0.75 or 1.5 μg mL$^{-1}$ Cy5-mRNA-LNP, or left untreated as control. Time-lapse image acquisition using a widefield microscope was started immediately after adding LNP to the culture medium. For each condition, two well separated positions were imaged in parallel, acquiring 30 z-planes with 300 nm step-size for 9–12 h with 10 min time interval. Imaging settings were kept identical between siRNA and mRNA experiments. The number of galectin-9 and LNP foci per cell was counted using CellProfiler. In brief, a median filter was first applied to the galectin-9 images, that were then subtracted from the raw images, to enhance small bright foci. Galectin-9 and LNP foci were then detected using a manual intensity and object size threshold defined for each channel. The number of foci was divided by the number of cells identified in the first frame of image acquisition. With galectin-9, the mean number of baseline foci per cell (typically present in cell nuclei) at start of image acquisition was subtracted from the total count per cell at following time-points. LNP intensity was calculated by quantifying the total fluorescence intensity of all LNP foci (sum) divided by the number of cells identified in the first frame. In Fig. 1b, the number of cells analyzed were 101, 105, 117, 106 and 124, 110, 120, 102 cells pooled from two independent experiments for each RNA modality, with 4, 20, 50, 100 nM siRNA- and 0.06, 0.30, 0.75, 1.50 μg mL$^{-1}$ mRNA-LNPs, respectively.

### siGFP-MC3, siGFP BODIPY-MC3 and mEGFP-MC3 LNP dose-response

For siGFP-MC3 LNP dose-response characterization, HeLa-d1-eGFP cells were plated in a 48-well plate at $3 \times 10^4$ cells per well. The following day, cells were treated with AF647-siGFP-MC3 LNPs at concentrations ranging from 7.8 pM to 128 nM prepared by 4-fold serial

dilution. Cells were treated with LNPs for 24 h, washed once with PBS and harvested for flow cytometry analysis of EGFP fluorescence as described below.

For comparing siGFP-MC3 and siGFP BODIPY-MC3 LNP dose-response, HeLa-d1-eGFP cells were plated in a 12-well plate at $1 \times 10^5$ cells per well. Cells were treated with 2, 10 or 50 nM AF647-siGFP-MC3 or AF647-siGFP BODIPY-MC3 LNPs for 24 h, followed by analysis by RT-qPCR as described below.

For mEGFP-MC3 LNP dose-response characterization, HeLa WT cells were plated in a 48-well plate at $3 \times 10^4$ cells per well. The following day, cells were treated with Cy5-mEGFP-MC3 LNPs at concentrations ranging from 0.09 to 1.5 µg mL$^{-1}$ prepared by 2-fold serial dilution. After 24 h, cells were harvested for flow cytometry analysis of EGFP fluorescence. In all cases, control cells were treated with imaging medium supplemented with rApoE.

### Endosomal release of LNP-RNA

HeLa-Gal9-YFP or HeLa cells transiently expressing mRuby-galectin-8 were continuously incubated with 4, 10, 20 or 50 nM MC3-siRNA-AF647 or 0.75 µg mL$^{-1}$ MC3-mEGFP-Cy5 as indicated. At start of galectin-9 or CHMP2A response (typically after 1–2 h), images were acquired using a widefield microscope. Typically, 4–10 well separated positions in a microscopy chamber slide well were imaged sequentially, acquiring 20 z-planes with 500 nm step-size for 100–200 time-points with 2.2–3.0 s time interval. Imaging settings were kept identical when acquiring all siRNA and mRNA channels. The galectin-9 or CHMP2A and RNA channels were acquired sequentially for each z-plane to minimize in-plane motion artifacts. Images were processed as described below before identifying events with de novo recruitment of galectin-9 or CHMP2A to cytoplasmic foci in Fiji. Foci were evaluated for presence of RNA. RNA$^+$ endosomes with no or only brief interaction with or juxtaposition to other RNA-containing objects, were manually tracked for at least 5 time-points before and/or after detectable galectin recruitment, using the Manual Tracking plugin. For CHMP2A expressing cells, only RNA$^+$ endosomes that were galectin-8$^-$ at the time of CHMP2A recruitment were included. Tracking coordinates were exported and used with an in-house developed MATLAB program for fluorescence quantification. An object mask with 6-pixel (620 nm) diameter (x and y) and 1 z-plane depth was fitted in x, y and z to locate the maximum RNA intensity of the tracked vesicle, before calculating the mean object intensity within the mask. Local fluorescence background was estimated as the median pixel intensity in a circular area 8 pixels (~825 nm) outside the fitted object mask. A local background rolling average was calculated over 11 frames and subtracted from the vesicle intensity measurement. Traces were aligned in time so that $t = 0$ corresponds to the first frame with detectable galectin recruitment. RNA measurements were normalized to the mean vesicle intensity ~3–60 s before detectable galectin recruitment. Galectin and CHMP2A traces were normalized to the maximum value. Vesicles showing fast initial RNA fluorescence decrease were identified as having a mean intensity between ~2.2–30 s after galectin recruitment below one standard deviation of the mean intensity calculated ~2.2–30 s before galectin recruitment. To correct for RNA signal decrease due to photobleaching, in experiment datasets, representative galectin-9$^-$ mRNA- and siRNA-LNPs were identified, tracked and measured during ~4.5 min using the method described above. The intensity values of individual vesicles were normalized to the first measurement in each series, before fitting an exponential function used as a reference for bleach correction. Photobleaching and corresponding correction was determined for all evaluated concentration of AF647-siRNA-LNP and Cy5-mRNA-LNP.

### Endosomal compartments damaged by LNPs

HeLa-Gal9-YFP cells with transient expression of mScarlet- or mCherry-tagged endosomal markers were prepared as described above. Cells

were continuously incubated with 50 nM AF647-siRNA-LNP or 0.75 µg mL$^{-1}$ Cy5-mRNA-LNP during experiments and imaging. At start of galectin-9 response (typically after 1–2 h), images were acquired using a widefield microscope, using the same general strategy and parameters as described above, but with a three-way image splitter to acquire channels partly simultaneous with complete spectral separation. All events with de novo recruitment of galectin-9 to RNA$^+$ intracellular compartments were identified and manually tracked as outlined above. Tracking was solely based on the RNA signal, quantifying the fluorescence intensity of all channels in the object mask. For each event, the presence or absence of endosomal marker on the damaged vesicle was evaluated manually. Compartment marker intensity values were corrected for local background as described above and normalized to the 90th percentile marker intensity per cell (representing a typical marker-labeled object). The 90th percentile intensity of compartment marker was calculated per cell at the beginning and end of each acquisition, and a linear regression model was fitted and used for time-dependent normalization. For quantitative event scoring, the compartment marker intensity of vesicles was assessed in an 11 s time-interval (five measurements) centered around the start of galectin recruitment. Vesicles with a normalized mean marker intensity >0.5 during the assessment interval were classified as marker positive.

### Colocalization of siRNA-LNPs and early or late endosomal compartments

HeLa-CD63-GFP cells or HeLa WT cells transiently expressing GFP-tagged Rab5 endosomal marker were prepared as previously described, and stained with 0.5 µg/mL Hoechst 33342 nuclear stain (Thermo Fisher Scientific) for 40 min. Cells were then continuously incubated with 10 nM or 50 nM AF647-siRNA-LNP, or left untreated as control. Time-lapse image acquisition using a LSM980 confocal microscope was started immediately after adding LNPs to the culture medium. For each condition, 3 well-separated positions were imaged in parallel, acquiring 3 z-planes with 2 µm step-size for 8 h with 5 min time interval. Images were denoised with LSM Plus (Zeiss) and exported as single plane images and as maximum intensity projection images.

The number of endosomal compartments and the colocalization between LNP foci and compartment markers were quantified using CellProfiler. Unless otherwise stated, the lowest z-plane, capturing the largest area of the cell, was used for all described steps and analyses. In brief, a speckle enhancement process was first applied to the LNP and compartment images. A median filtered image was then subtracted from the processed image to remove noise and obtain clear foci. Cells were segmented with watershed processing to analyze marker-positive cells, using maximum intensity projection images of nuclear fluorescence as a seed point and compartment marker fluorescence to delineate cell borders. Within the segmentation mask of each cell, LNP and compartment foci were identified using manually defined intensity and object size thresholds per channel. The number of compartment foci was normalized by the number of marker-positive cells per frame and to the average number of compartment foci per cell in the first frame of acquisition. Colocalization was assessed by resizing the LNP and compartment foci to circular mask with a diameter of 5 pixels, and subsequently calculating the number of LNP foci overlapping with compartment marker foci.

### CHMP2A response to RNA-LNP treatment

HeLa cells transiently expressing GFP-CHMP2A were continuously incubated with 100 nM AF647-siRNA-LNP or 1.5 µg mL$^{-1}$ Cy5-mRNA-LNP. Imaging medium supplemented with rApoE was used as control treatment. For each condition, a single position was selected for time-lapse imaging using widefield microscopy, acquiring 20 z-planes with 500 nm step-size every 10 or 30 min for 4 h. Maximum intensity projections were created in Fiji, and the number of CHMP2A foci per cell was quantified at the start of treatment and at 1, 2 and 3 h through

blinded manual inspection. The CHMP2A foci identified at 1 and 3 h of treatment were also manually classified to be either positive or negative for AF647-siRNA or Cy5-mRNA through visual assessment.

### ESCRT knockdown, LNP internalization and galectin-9 response

HeLa-Gal9-YFP cells were transfected with siRNA targeting Alix, Tsg101 or CHMP6 (0.4 µM each) or negative control siRNA (0.8 or 1.2 µM) by electroporation as described above. Approximately $3 \times 10^4$ cells were immediately seeded per well into an 8-well Chamber Slide (galectin-9 response), and ~$2 \times 10^5$ cells were plated per well in a 6-well plate (mRNA knockdown). Alternatively, electroporated cells were immediately seeded at ~$3 \times 10^4$ cells per well in a 48-well plate (LNP internalization). All subsequent experiments were performed 24 h after transfection.

For evaluating target knockdown, cells cultured in the 6-well plate were harvested for mRNA quantification using RT-qPCR as described below.

For evaluating galectin-9 response to LNP treatment, cell culture medium was changed to imaging medium supplemented with rApoE and $0.12\,\mu g\,mL^{-1}$ Hoechst 33342 before transferring cells to a heated microscope stage top incubator. Immediately prior to start of imaging, $0.5\,\mu g\,mL^{-1}$ mEGFP-MC3 LNPs were added to the respective well. For ESCRT knockdown and control cells, 7 positions were imaged consecutively per condition at each time-point, acquiring 6 z-planes per position with 1-µm interval every 10 min for 18 h using a LSM980 confocal microscope. Images were denoised using LSM Plus (Zeiss) before further processing. Galectin-9 foci were quantified per cell from maximum intensity projections of z-stacks using Cell Profiler. For each experiment, galectin-9 response was normalized to the maximum number of foci detected over the course of the experiment in cells receiving negative control siRNA.

For evaluating LNP internalization, $0.5\,\mu g\,mL^{-1}$ Cy5-mRNA-MC3 and 20 nM AF647-siGFP-MC3 LNPs were prepared in imaging medium supplemented with rApoE. Cells were washed once with PBS followed by continuously LNP treatment for 2, 4 or 6 h. Imaging medium with rApoE was used as negative control. At the end of experiment, cells were washed once with PBS and then harvested for flow cytometry analysis of AF617 or Cy5 fluorescence as described below. Background fluorescence intensity was estimated using the negative control sample and was subtracted from LNP treated samples.

### LNP disintegration

HeLa cells transiently expressing GFP-EEA1, GFP-Rab5 or GFP-CD63 were plated in microscopy slides as described. The cells were transferred to a preheated incubation chamber of the LSM710 confocal microscope and treated with 100 nM AF647-siRNA LNPs or $1.5\,\mu g\,mL^{-1}$ Cy5-mRNA LNPs. Cells expressing GFP-EEA1 were treated with the LNPs for approximately 15–75 min, GFP-Rab5 cells for 30–90 min and GFP-CD63 cells for 45–120 min and imaged live during LNP incubation. Single z-plane images were acquired, as at least 30 images per experiment, of multiple cells from different areas in the well. The image size was 1024 × 1024 pixels, with a pixel size of 9 nm, except for when imaging a full cell per field of view, when a pixel size of 20 nm was used. To measure the size of single LNPs, AF647-siRNA or Cy5-mRNA LNPs (10 nM or $0.15\,\mu g\,mL^{-1}$ respectively) was diluted in sterile-filtered deionized water and added to microscopy chamber slides. LNPs were imaged immediately, as single z-plane images were acquired using the same laser parameters and pixel size as when imaging the LNP treated HeLa cells. In Fiji, 120 × 120 pixels region of interests (ROIs) of single LNPs structures were manually selected and saved as new images. From images acquired of GFP-EEA1 expressing cells, LNPs colocalizing with EEA1+ structures were included. The full-width at half maximum (FWHM) of the LNPs were measured using a MATLAB script, by fitting a 2D gaussian curve to the LNP intensity[45].

To determine the frequency of endosomes with confined or dispersed luminal RNA signal in EEA1+, Rab5+ or CD63+ endosomes, a manual classification was performed. Firstly, the image with the compartment marker was viewed and endosomes with a clear lumen were marked to be include for further inspection, to confirm that endosomes were captured in the centermost focal plane. Next, the LNP appearance was classified into two subgroups, where the luminal RNA signal either filled the entire lumen (dispersed signal) or not (confined signal), as a measurement of LNP disintegration. A small subset of endosomes (<9% per condition) were discarded from analysis, where a clear classification could not be performed.

To evaluate the LNP disintegration in 3D, HeLa cells expressing GFP-CD63 or GFP-EEA1 were plated in microscopy slides as described and placed in a preheated incubation chamber of the VT-iSIM. The cells were treated with 100 nM AF647-siRNA LNPs and imaged 30-90 min or 100-130 min after LNP addition for EEA1 or CD63 expressing cells respectively, as typically 20-40 z-planes with a 100 nm step-size were acquired. Reference stacks of 0.1 µm TetraSpeck microspheres (Thermo Fischer Scientific), prepared on glass slides with glycerol, were acquired prior to the start of each experiment. The bead reference images were acquired with identical z step-size as in the experiments and were used as a reference for channel alignment in z. The image size was 2119 × 1963 pixels, with a pixel size of 46 nm. Images were deconvolved using Huygens Professional. Orthogonal views were computed with Fiji. Due to spherical aberration introduced by using an oil objective, there was apparent elongation in the z-axis[46]. To compensate for this, the xz and yz images of the orthogonal views were corrected using a scaling factor of 0.667.

### Spectrophotometric measurement of LNP fluorescence

Naked AF647-siRNA, AF647-siRNA LNPs, AF647¼-siRNA LNPs, Cy5-siRNA LNPs, AF647-siRNA MC3-BODIPY LNPs, Cy5-mRNA LNPs and Cy5-mRNA BODIPY-MC3 LNPs were diluted in OptiMEM (Gibco) to a final concentration of 50 nM or $0.75\,\mu g\,mL^{-1}$ for the siRNA or mRNA respectively and divided into two tubes each. LNPs in one subset of tubes were dissociated with 1% Triton-X-100 (Sigma). The fluorescence was measured using a FLUOstar OPTIMA spectrophotometer (BMG LABTECH, Ortenberg, Germany) (excitation wavelengths; 485 nm and 590 nm, emission wavelengths; 520 nm and 660 nm) with FLUOstar OPTIMA software v.1.32 R2. Two technical replicates were used and the mean of the duplicates was calculated. Background fluorescence in OptiMEM was measured and subtracted before plotting the data.

### BODIPY-MC3 and RNA segregation

HeLa wild-type cells or HeLa cells transiently expressing mScarlet-Rab5 were seeded into microscopy chamber slides as previously described and placed in a preheated incubation chamber of the VT-iSIM. The cells were treated with 100 nM AF647-siRNA MC3-BODIPY LNPs or $1.5\,\mu g\,mL^{-1}$ Cy5-mRNA BODIPY-MC3 LNPs and imaged 1–4 h after LNP addition, as generally 5 z-planes with a 500 nm step-size per field of view were acquired. Laser power settings were determined by firstly imaging intact, single AF647-siRNA MC3-BODIPY or Cy5-mRNA BODIPY-MC3 LNPs prepared in diH$_2$O (10 nM or $0.15\,\mu g\,mL^{-1}$ for siRNA or mRNA respectively) in glass slides. Laser power and exposure times were adjusted so both BODIPY-MC3 and RNA were visible in the great majority of all observed structures. The following exciting laser lines and single bandpass emission filters were used for the respective channels: BODIPY (common emission spectrum): 488-nm laser, 530 ± 30 nm filter; redshifted BODIPY: 488-nm laser, 600 ± 50 nm filter; BODIPY-AF647 FRET: 488-nm laser, 690 ± 50 nm filter; AF647/Cy5: 640-nm laser, 690 ± 50 nm filter; mScarlet: 561-nm laser, 600 ± 50 nm filter. Images were deconvolved with Huygens Professional.

For each image, typically containing 1–4 cells per FOV, cell outlines and nuclei were manually delineated in Fiji and the ROIs were

saved as separate images. Next, LNP foci or structures were identified and segmented in each channel (RNA, free BODIPY-MC3, redshifted BODIPY-MC3 and FRET), in Cell Profiler using a custom pipeline. Segmentation was achieved by applying a maximum correlation thresholding (MCT), and the resulting images were exported as labeled uint16 images. Images were then analyzed by a custom MATLAB script. First, the centroid of all nuclei of analyzed cells were determined. For all channels, identified structures were then individually assessed, calculating the distance from the object center (centroid) to nucleus center and the colocalization of objects in the other channels. If an evaluated structured overlapped with a segmented object in a different channel, where the colocalizing object overlapped with >25% of its area, the objects were classified as colocalizing.

### Sub-endosomal localization of LNP and membrane damage

HeLa-Gal9-YFP cells were transfected with mScarlet-Rab5 as described above. HeLa wild-type cells were transfected with GFP-CHMP2A and mScarlet-EEA1, or GFP-CHMP2A and mScarlet-Rab5. Cells were plated in microscopy slides, and the following day, treated with 100 nM AF647¼-siRNA LNPs for 60 min (CHMP2A) or 90 min (galectin-9). Cells were then washed with PBS, fixed with 4% paraformaldehyde (PFA) for 10 min in room temperature and washed three times with PBS. Fixed cells were kept in PBS after last wash and imaged within 2 days using Airyscan confocal microscopy. Cells were selected for imaging based on endosomal marker expression level.

To study the sub-endosomal localization of the damage marker in relation to the LNPs, a custom analysis pipeline was developed. Firstly, from the images acquired, endosomes where a clear lumen could be distinguished were identified. In these, the ones that were positive for both the endosomal membrane damage markers (GFP-CHMP2A or YFP-galectin-9) and AF647¼-siRNA were selected for further analysis. Furthermore, since the localization within single endosomes was desired, clustered endosomes were excluded from analysis. In Fiji, 130 × 130 pixel ROIs of the relevant endosomes were manually created and saved as new images. In these images, the endosomes were categorized into two different categories, one where the LNP and/or damage marker exhibited a polarized localization and one where they were localized in the endosome center. The endosomes where the LNP and/or damage marker were centralized were excluded from analysis. Using a custom MATLAB script, the endosomes were masked and the center of mass of the endosome was identified. This step was manually inspected for each endosome. Next, 1–3 local intensity maxima within the endosomal mask were identified in both the LNP channel and the damage marker channel. A vector from the center of mass to the local maxima was then defined for each maximum. The local maxima of the two channels (damage marker and LNP) were then manually paired, and 1–2 paired maxima were included for further analysis. For these, the angular difference and magnitude difference (in nm) between the vectors were calculated. Precision above the diffraction limited resolution of the imaging system was obtained analogous to ref. 47.

Furthermore, HeLa cells were transfected with mCherry-galectin-9 and plated in microscopy slides as described previously. The cells were transferred to a preheated incubation chamber of the VT-iSIM and treated with 100 nM AF647-siRNA BODIPY-MC3 LNPs. Approximately 90–180 min after LNP-addition, the cells were imaged, as typically 20 z-planes with a 250 nm step-size were acquired. The following exciting laser lines and single bandpass emission filters were used for the respective channels: BODIPY: 488-nm laser, 530 ± 30 nm filter; mCherry: 561-nm laser, 600 ± 50 nm filter; AF647: 640-nm laser, 690 ± 50 nm filter. Images were deconvolved with Huygens Professional and inspected for endosomes with membrane-bound BODIPY-MC3 and galectin-9 recruitment.

### Flow cytometry

At the end of experiments, cells were washed once with PBS, detached through trypsinization and transferred to 12 × 75 mm polystyrene FACS tubes or a 96-well V-bottomed microplate. Cells were centrifuged at 400 × $g$ for 5 min, the supernatant was removed and cells were resuspended in 0.5% Bovine Serum Albumin (BSA, Sigma) in PBS. Cells were centrifuged again and resuspended in 0.5% BSA in PBS. Cells were analyzed using an Accuri C6 Flow Cytometer (Becton Dickinson, Franklin Lakes, NJ, USA) with BD Accuri C6 Software v.1.0.264.21. Live cells were gated in side scatter/forward scatter plots and the median fluorescence intensity was measured. EGFP fluorescence was measured in the FL1 (533 ± 30 nm) channel. AF647 and Cy5 fluorescence was measured in the FL4 (675 ± 25 nm) channel. For EGFP quantifications with HeLa-d1-eGFP, HeLa WT FL1 fluorescence was subtracted to remove background fluorescence. For most experiments, the mean of two technical replicates was calculated. When applicable, data was normalized as described in Figure legends.

### Reverse-transcription quantitative PCR

At the end of experiments, cells were washed once with PBS and RNA extraction was performed using GenElute Mammalian Total RNA Miniprep Kit (Sigma), according to manufacturer's protocol. Next, complementary DNA was synthesized with SuperScript III First-Strand Synthesis System (Sigma) with random hexamers, on a MasterCycler EpGradient 5341 thermal cycler (Eppendorf AG, Hamburg, Germany). Approximately 100 ng total RNA was used for cDNA synthesis. For real-time qPCR, SYBR Green JumpStart Taq ReadyMix (Sigma) was used. All primers were used at 200 nM. Reactions were performed in MicroAmp Fast 0.1 mL 96-well Reaction Plates (Applied Biosystems, Foster City, CA, USA) on a StepOnePlus Real-Time PCR System. Data was analyzed using StepOne Software v2.3. GAPDH was used as housekeeping gene, and mRNA expression fold change compared to the control was calculated using the $\Delta\Delta C_T$ method.

### Microscopy

Four imaging platforms were used, as designated in the text. Relevant acquisition parameters of experiments are given in the corresponding methods section. All platforms have heated stage-top incubators and heating inserts (Pecon), Definite focus, $CO_2$ control systems, and are operating at 37 °C and 5% $CO_2$ during live-cell experiments.

LSM710 Airyscan confocal platform: Inverted Axio Oberver Z.1 LSM 710 confocal laser scanning microscope with Airyscan detector (Carl Zeiss AG, Oberkochen, Germany), equipped with a Plan-Apochromat ×63/1.40 oil-immersion objective lens (Zeiss), diode laser 405 nm, Lasos Argon laser 458/488/514 nm, DPSS 561 nm and HeNe laser 633 nm. The system was equipped with a 488/651/633 main beam splitter and BP 495–550 + LP 570 nm, BP 420–480 + BP 495–620, BP 570–620 + LP 645 emission filters (all Zeiss). The BP 495–550 + LP 570 nm emission filter was used for all two-color imaging. The BP 495–550 + LP 570 nm was used for GFP/YFP, BP 420–480 + BP 495–620 was used for mScarlet and BP 570–620 + LP 645 for AF647 for three-color imaging. The Airyscan detector was used in super-resolution mode for all channels in all experiments. Image size was 1024 × 1024 pixels, with a pixel size of 9 nm, if not stated otherwise. The system operates under ZEN 2.1 (black).

LSM980 confocal platform: Inverted Axio Observer 7 LSM980 confocal laser scanning microscope (Zeiss), equipped with a 32-channel GaAsP spectral PMT detector, C Plan-Apochromat ×63/1.40 oil-immersion objective lens and Piezo Z stage (Wienecke & Sinske GmbH, Gleichen-Bremke, Germany). Images were acquired in confocal mode with ×0.6 scan zoom using a 405-nm exciting diode laser with 405/730-nm main beam splitter and a 488-nm exciting diode laser with a 488/639-nm main beam splitter. Image size was 2048 × 2048 pixels, with a pixel size of 0.11 μm. The system operates under ZEN 3.8.2 (blue).

Widefield platform: Inverted AxioOberver Z.1 microscope equipped with a Plan-Apochromat ×63/1.40 oil-immersion objective lens and Colibri 7 solid state LED light source (Zeiss), MS2000 XY Piezo *Z* stage (Applied Scientific Instrumentation, Euguene, OR, USA), and an ORCA-Flash4.0 V3 Digital CMOS camera (Hamamatsu Photonics, Hamamatsu City, Japan). The system was equipped with a three-way image splitter (Optosplit III, Cairn Research Ltd, Faversham, Kent, UK) using a turret quad band beam splitter and excitation filter (89402 ET – 391 ± 32/479 ± 33/554 ± 24/638 ± 31, Chroma Technology Corp, Bellows Falls, VT, USA) and a splitter cube containing a 565 nm LP (T565lpxr) and 635 nm LP (T635lpxr) beam splitter and separate 519 ± 26, 595 ± 33 and 697 ± 60 nm emission filters (ET519/26 m, ET595/33 m, ET697/60 m, all from Chroma Technology). For experiments with two channel acquisitions, the splitter was used in bypass (single channel) mode, with a multiband turret filter set (90 HE LED (E), Zeiss) (Ex 385, 475, 555, 630 nm, QBS 405, 493, 575, 653 nm, QBP 425 ± 30, 514 ± 30, 592 ± 30, 709 ± 100 nm) or a bandpass turret filter (38 HE, Zeiss) (Ex 470 nm, beam splitter FT 495 nm, BP 425 ± 50 nm) and sequential channel acquisitions. For experiments with three channel acquisitions, simultaneous acquisition of GFP/YFP and AF647/Cy5 was performed by illumination with the 475-nm and 630-nm LED, followed by mCherry/mScarlet acquisition using the 567-nm LED with the 555 ± 30 nm filter active. The system was used in fast acquisition mode (camera, illumination and stage) using a SVB 1 signal distribution box (Zeiss) and µCon HS Trigger board (PCIxPress). The system operates under ZEN 2.3 (blue). Channels were acquired sequentially for each *z*-plane, to reduce latency between channels in the same *z*-plane.

VT-iSIM platform: Inverted Axio Oberver Z.1 microscope (Zeiss), equipped with a Plan-Apochromat ×100/1.46 oil-immersion objective lens (Zeiss) and MS2000 XY Piezo Z stage (Applied Scientific). The system includes a VT-iSIM Super Resolution Confocal Head, a VT-LMM Laser Engine (488 nm, 561 nm and 640 nm lines were used), VT Dual Cam image splitter (all supplied by VisiTech International Ltd, Silverbriar, Sunderland, UK), and two ORCA-Quest qCMOS cameras (Hamamatsu). Either a 577-nm LP or 647-nm LP dichroic beam splitter (ZT568rdc or T647lpxr respectively) was used in the Dual Cam splitter. The following emission filters were used: Camera 1 (SP): 530 ± 30; Camera 2 (LP): 600 ± 50, 690 ± 50 (ET530/30 m, ET600/50 m, ET690/50 m, all filters from Chroma Technology). The system operates under VisiView Version 6.0.0.16. (Visitron Systems GmbH, Puchheim, Germany).

**Widefield microscopy image processing**
A previously described MATLAB program[32] was adapted for image processing and analysis. In brief, images were exported as raw 16-bit tiff-files. With multichannel (image splitter) exposures, channels were separated into individual tiff-files followed by axial and lateral channel alignment. A reference dataset of 0.1 µm TetraSpeck microspheres (Thermo Fischer Scientific) prepared on a glass slide in glycerol, were acquired before start of every experiment and used as a reference for channel alignment (image splitter datasets) and chromatic aberration correction (all datasets). The bead reference dataset was acquired with identical filter settings as the corresponding experiment, but with 260 nm *z*-interval. Correct channel alignment was determined with 1 pixel (lateral) and 1 *z*-interval (axial) accuracy by image translation to maximize bead overlap in all channels. Any remaining channel misalignment due to chromatic aberration introduced in the optical pathway (for both image splitter and single-channel datasets) was determined by calculating the distance between weighted centroids of beads between channels. A linear model was fitted to the measurements (for *x* and *y* separately) and used to correct lateral chromatic aberration with 1 pixel accuracy before further analysis or visualization. Images were deconvolved using Huygens Professional and empirical point spread functions before analysis.

**Software**
MATLAB 2019b was used for data analysis as described above. CellProfiler 2.1.1 with customized pipelines was used for galectin and LNP puncta quantifications and segmentation of cells and intracellular objects. GraphPad Prism 10 for macOS Version 10.0.2 (GraphPad Software, Boston, Massachusetts USA) was used to create figures and perform statistical testing. Fiji 1.52i was used for analysis and visualization of microscopy images. Brightness and contrast settings were linearly adjusted and kept identical for images intended for comparison unless otherwise stated in Figure legends. Final figures were composed in Adobe Illustrator Version 27.9 for macOS. Huygens Professional for Windows Version 17.04 and 23.04 (Scientific Volume Imaging B.V., Hilversum, Netherlands) was used for widefield and iSIM microscopy image deconvolution, respectively.

**Reporting summary**
Further information on research design is available in the Nature Portfolio Reporting Summary linked to this article.

## Data availability
All data supporting the graphs in this paper is available in the Source Data file. Source data is available for Figs. 1b, d–f, 2c, d, f–h, 3c–e, 4c–e, 5a, c, 6b, 7d, c and Supplementary Figs. S1a–e, S2d, S3a–c, S4a–c, S5a, S5b, S6a–e, S7a–c, S8b, S9a–d. Underlying imaging files and all other raw data files are available from the corresponding author upon request (due to the size of this material). Source data are provided with this paper.

## Code availability
Custom MATLAB code used for data analysis in this paper is available at https://github.com/hdurietz/BarriersToLNPdelivery.

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

## Acknowledgements

We are grateful to Victor Spelling and Fritz Schweikart (AstraZeneca R&D Gothenburg) for help with purification and characterization of the BODIPY-MC3 ionizable lipid. We are also grateful to Martin Sundwall (Lund University) for the cloning of Rab5-mScarlet and to Rebecka Cattani for assistance with ESCRT quantifications. This work was supported by grants to A.W. from the Swedish Society for Medical Research (SSMF), the Gunnar Nilsson Cancer Foundation, the Mrs. Berta Kamprad Foundations (FBKS-2020-33-308 and FBKS-2022-40-422), the Winkler's Foundation, the Governmental funding of clinical research within the National Health Services (A.L.F., YF2023), the Swedish Research Council (VR, 2020-02647), the Wallenberg Center for Molecular Medicine, Lund University, and the Knut and Alice Wallenberg foundation.

## Author contributions

A.W. conceived the study and supported funding. J.M.J., H.D.R., H.H. and A.W. designed the experiments, analyzed and interpreted the data, and created figures. J.M.J., H.D.R., and H.H. performed most of the experiments. H.C.E. performed some of the experiments. E.O.B. formulated LNPs. A.P. synthesized BODIPY-MC3. S.H. analyzed BODIPY-MC3 integrity. H.D.R. designed software tools for image processing. P.N. assisted in fluorescent protein design. L.L. conceived BODIPY-MC3 and contributed to experimental design, material selection, analysis strategies and interpretation of data. J.M.J., H.D.R., H.H. and A.W. wrote the paper with input from all authors.

## Funding

## Competing interests
E.O.B., A.P., S.H. and L.L. were employed by AstraZeneca R&D at the time of this work. The remaining authors declare no competing interests.
