## [Transparent Peer Review file · Nature Communications]

Cellular and biophysical barriers to lipid nanoparticle mediated delivery of RNA to the cytosol

Corresponding Author: Dr Anders Wittrup

Version 0:

Reviewer comments:

Reviewer #1

(Remarks to the Author)

The study by Wittrup and colleagues uses high resolution live-cell microscopy to monitor the trafficking and escape of lipid nanoparticles containing mRNA and siRNA, thereby assessing barriers that limit the efficiency of lipid nanoparticle-mediated RNA delivery from endosomes into the cytosol. This is a vexing problem in biotechnology and the authors are correct in saying that it has not received much attention in cell biology. The results – while more descriptive than mechanistic – establish that cargo release is rarely complete and, unexpectedly, that lipid and RNA cargo segregate from each other as lipid nanoparticles disintegrate during endosomal maturation potentially contributing to localized membrane damage. They connect these results to earlier studies of membrane damage and repair by showing a small increase in galectin accumulation in cells depleted of multiple ESCRTs and exposed to mRNA-LNP. The technical advance of quantitatively imaging both fluorescent mRNA/siRNA cargo in parallel with ionizable lipid can help to stimulate future mechanistic studies in this area.

Issue to address:

-The experiment in Fig. 2E assessing the effect of ESCRT depletion on mRNA-LNP induced membrane damage (as measured by accumulation of galectin puncta) needs further development. Why are three ESCRTs depleted (especially why include CHMP6) and why is the imaging data supporting this graph not shown? This is important as it represents really the only connection between these descriptive studies and previous mechanistic cell biology.

Reviewer #2

(Remarks to the Author)

The use of nucleic acids has huge potential in treatment of disease yet there are many roadblocks for maximising the potential of these cargos. This study looks at one of the key barriers to delivery which is effective endosomal escape well known to be low (2-5%). Importantly, the mechanisms to understand this process are also poorly understood. In this paper the authors look to design a comprehensive study to understand the endosomal escape behaviour of LNPs with siRNA and mRNA cargo. I thought this study was well done but I think it leaves too many questions to add substantially to the literature. For instance, the hit rate for mRNA and siRNA highlighted in Figure 1e as well as the different LNP behaviour in Figure 4. I would have liked to see some model studies used to attempt to come up with a rationale for the large differences between LNPs containing siRNA and mRNA. The authors talked about segregation in their discussion of this phenomena but they did not do this experiment with mRNA (only siRNA in Figure 5). It seems like an obvious experiment to improve their study. In summary, while I think this an important area, I am not convinced that this paper adds a lot of concrete understanding to the area beyond visualising the processes of galectin and ESCRT recruitment correlated with RNA release, thus I would suggest publication only after major revision. I have added some additional points for clarification below.

- I would say the LNP internalisation looks similar from 20 nM through to 100 nM (Figure S1a) with the green curve (20nM) tracking higher than the other two. Thus, why is the galectin-9 foci so much lower in the 20 nM case (Figure 1b)? In addition, does the amount of foci correlate to membrane disruption and thus escape. The results on the ESCRT seem to suggest it is as the authors saw 20% increase in foci with ESCRT knockdown. I think the importance of galectin-9 foci could be made clearer?

- The authors showed that there was an impact on RNA load for siRNA systems but none for mRNA (line 191-194), why would there be a difference?

- The paper does not really discuss the intensity of the galectin-9 recruitment which is interesting as it varies quite widely, I think this would be a helpful addition?
- I had a question about figure 3a/b, I think the pattern of the recruitment of galectin and EEA1 looks different in the two types of RNA, in siRNA you have lost the signal of EEA1 when the galectin starts to increase but in the mRNA the EEA1 stays consistent long after this time. Is this representative of the rest of the data and if so, what does it mean?
- Figure 4 clearly shows a change in the siRNA distribution, but I was not sure if these were really single LNPs, the cross-section suggests these are 300-500 nm, this is large? I would also say the CD63 images do not look like a diffuse breakdown of one or two particles as the fluorescence is not uniform, it still looks punctate in multiple cases (Figure 4b), how do we know this is not the result of more LNPs in the endosomes? I also think the same cross-section data should be added for the mRNA data in Figure 4 for comparison.
- Why wasn't the FRET study done with mRNA and siRNA (Figure 5)? I think this is a limitation with the manuscript as everything to this point has been done with point and they show clear differences, this experiment might have offered some explanation for these findings? I also think the evidence of separate BODIPY and siRNA is pretty weak (I think it is only demonstrated using one small image in Figure 5b). I don't find this convincing. The authors claim that the addition of BODIPY into the membrane supports it plays a role in membrane disruption (line 360-361), while I think that is likely true I do not think this data proves it. Can the authors please explain this point more clearly?

Reviewer #3

(Remarks to the Author)

In the manuscript titled "Cellular and biophysical barriers to lipid nanoparticle mediated delivery of RNA to the cytosol" authors use microscopy to unravel various processes involved post the endocytosis of RNA-LNPs. By tracking single vesicles containing LNPs, the authors describe the damage marked by galectin recruitment. Further, the co-localization of LNPs with different endosomal compartments (EEA1, Rab5, CD63) is quantified and the data shows the highest load of RNA in early endosome vesicles marked by Rab5. Using super-resolution microscopy, the authors image the disintegration of LNPs to be happening in late endosome vesicles marked by CD63. Finally, the recruitment of gal9 to Rab5/EEA-1 vesicles containing siRNA-LNPs is microscopically evaluated to determine the points of damage to the endosomal membrane. Overall, the study employs various microscopic tools to gain mechanistic insight into the process of endosomal escape. The observations reported are important to the burgeoning field of LNP-based therapeutics. While the manuscript is meticulously written and microscopic data is detailed and rigorous, there are some concerns with the interpretation and robustness of the data. Insight into the bigger picture and how the observations might affect the choice of the vehicle and the payload is missing.

1. In Figure 1f, the decrease in the fluorescence intensity upon Gal9 recruitment is interpreted as an event of endosomal release of the RNA. Why is the drop in fluorescence interpreted as the release of mRNA? Have the authors considered the possibility of RNA being degraded inside the vesicles? Additionally, in case there is an actual endosomal escape happening, the decrease in the fluorescence inside the vesicles will lead to a corresponding increase in the fluorescence outside the vesicles (i.e., cytoplasm); can the authors observe this event?
2. Throughout the manuscript, information should be provided on the number of cells being analyzed for each data set presented in the graphs.
3. Fig1e: Do the authors have a hypothesis on why only a fraction of gal9+ vesicles contain the mRNA? What was the time point for data analysis in Fig 1e post-LNP addition? Did this fraction change with time post-LNP addition?
4. In Fig2, the authors report the no. of foci of CHMP2A analyzed. It is important to also provide information on the no. of cells being evaluated. In addition to this, it is more informative to the readers if images of more than one cell are provided in the main figures.
5. On pg10, line 256, the authors write "EEA1 intensity generally peaked before endosome damage, while Rab5 was more consistent over the observed interval". It will be good to discuss why this is happening and if it relates to the signature of the compartment where the endosomal release is happening. Following this, if the endosomal release is occurring from Rab5/EEA1 +/- vesicles, why is the RNA load similar in all the endosomal compartments (supp fig 4b)? Further, with the increase in the time of incubation of mRNA/siRNA LNPs with the cells, is there a change in the proportion/distribution of endosomal compartments-mRNA?
6. Colocalization of gal9, EEA1 and RNA is an (fig 3A) observation. More proof is required to prove that Rab5 is the compartment where endosomal release occurs. Will the endosome escape (mRNA expression) be affected if you knock down/downregulate Rab5?
7. Based on data presented in Fig4., the disintegration of LNPs occurs in the late endosome stages and not in the early stages as depicted in the microscopic images. Since the mRNA release can occur only after the LNPs start to disperse, one would expect the release to occur from CD63+ compartments/stages later than Rab5+ compartments. It would be great if the authors could shed some light on this and correlate this data with the observations in the previous figure.
8. Why have the authors used ApoE in the cell culture media? Since the media is supplemented with FBS, the proteins present in the serum will enable the endocytosis of LNPs.
9. In Fig3: Since the mRNA expression/siRNA activity increases with increasing concentration of RNA, it will be good to see if increasing RNA conc. in the medium leads to an increase in the number of different endosomal compartment vesicles.
10. Pg14 lines 347 and 346, it would be informative to show the colocalization of Bodipy-MC3 LNPs with gal 9 vesicles.
11. In suppl. Fig7., the label on the right panel of the image should be corrected to CD63+ vesicles. Interestingly, in this panel, only a few CD63+ vesicles appear to contain the dispersed RNA and the LNPs outside CD63+ vesicles appear diffused/disintegrated. At this stage, are the LNPs present in some other vesicle types eg., those marked by Rab7/Lamp1?
12. In Fig. 4, confocal microscopy is used to determine the localization of siRNA within different endosomal compartments. Although ingenious and interesting, this method does not enable, without a z-stack, to appreciate siRNA distribution in the entirety of the vesicle, and thus, to completely trust the resulting observations.

13. In Fig. 2b, an increase of CHMP2A foci in the presence of RNA cargo is reported, suggesting the recruitment of membrane repair systems. Considering the images displayed in Fig. 2. a, it would be interesting to exploit them further to quantify the colocalization of such foci with fluorescent RNA precisely.

14. Finally, authors use MC3 as an ionizable lipid for all their studies. This is not the best option for mRNA delivery. It will be interesting to see how the observations change with the change of ionizable lipids for example with ALC-0315 9Pfizer/bnt) or SM-102 of Moderna.

Version 1:

Reviewer comments:

Reviewer #2

(Remarks to the Author)

I have reviewed the responses of the authors to my comments on their manuscript and I believe they have sufficiently addressed all points with greater discussion on some of the key concepts such as why they see different hit rates between siRNA and mRNA systems as well as some further data supporting different behaviour of these systems. Thus, I am happy to support publication in this revised form.

Reviewer #3

(Remarks to the Author)

The authors have nicely revised the manuscript and it is now suitable for publication.

- Dan Peer

Rebuttal letter for “Cellular and biophysical barriers to lipid nanoparticle mediated delivery of RNA to the cytosol”.

2025-02-05

Dear reviewers,

We sincerely thank you for taking the time to review our manuscript, “*Cellular and biophysical barriers to lipid nanoparticle-mediated delivery of RNA to the cytosol.*” We greatly appreciate the insightful comments and constructive suggestions provided by the reviewers, which have helped us to significantly enhance the quality and clarity of the manuscript. We have tried to address all the concerns and recommendations raised and believe that the revised version reflects these improvements. We look forward to your feedback on the updated manuscript and hope it meets the expectations of the reviewers and the editorial team.

REVIEWER COMMENTS

Reviewer #1 (Remarks to the Author):

The study by Wittrup and colleagues uses high resolution live-cell microscopy to monitor the trafficking and escape of lipid nanoparticles containing mRNA and siRNA, thereby assessing barriers that limit the efficiency of lipid nanoparticle-mediated RNA delivery from endosomes into the cytosol. This is a vexing problem in biotechnology and the authors are correct in saying that it has not received much attention in cell biology. The results – while more descriptive than mechanistic – establish that cargo release is rarely complete and, unexpectedly, that lipid and RNA cargo segregate from each other as lipid nanoparticles disintegrate during endosomal maturation potentially contributing to localized membrane damage. They connect these results to earlier studies of membrane damage and repair by showing a small increase in galectin accumulation in cells depleted of multiple ESCRTs and exposed to mRNA-LNP. The technical advance of quantitatively imaging both fluorescent mRNA/siRNA cargo in parallel with ionizable lipid can help to stimulate future mechanistic studies in this area.

Issue to address:

1.1 The experiment in Fig. 2E assessing the effect of ESCRT depletion on mRNA-LNP induced membrane damage (as measured by accumulation of galectin puncta) needs further development. Why are three ESCRTs depleted (especially why include CHMP6) and why is the imaging data supporting this graph not shown? This is important as it represents really the only connection between these descriptive studies and previous mechanistic cell biology.

We welcome the reviewer’s suggestions regarding our ESCRT data. We have addressed these concerns by presenting additional experiments that have evaluated individual and different combinations ESCRT components with respect to the induction on galectin-9 foci during mRNA-LNP treatment, shown in Fig. 2g. Specifically, we investigated knockdown of

CHMP6 alone, dual knockdown of CHMP6 + TSG101 and Alix + TSG101. Compared to simultaneous depletion of CHMP6 + TSG101 + Alix as we presented in the original submission (Fig. 2h), dual knockdown of CHMP6 + TSG101 increased the number of galectin-9 foci during LNP treatment in approximately equal extent (~20%), while knockdown of Alix + TSG101 or CHMP6 alone made no difference when compared to control.

The inclusion of CHMP6 is motivated by its documented role in endolysosomal escape and membrane integrity, as shown by Chen et al., who demonstrated that compromised ESCRT function facilitates endolysosomal escape of pathological tau seeds and aggregation. This additional reference is now cited to underscore the specific role of CHMP6 in mediating endosomal integrity and its relevance to our observations.

We have included representative examples of imaging data used for the quantifications of galectin-9 foci after ESCRT knockdown in Supplementary figure 5c.

Reviewer #2 (Remarks to the Author):

The use of nucleic acids has huge potential in treatment of disease yet there are many roadblocks for maximising the potential of these cargos. This study looks at one of the key barriers to delivery which is effective endosomal escape well known to be low (2-5%). Importantly, the mechanisms to understand this process are also poorly understood. In this paper the authors look to design a comprehensive study to understand the endosomal escape behaviour of LNPs with siRNA and mRNA cargo. I thought this study was well done but I think it leaves too many questions to add substantially to the literature.

- 2.1 For instance, the hit rate for mRNA and siRNA highlighted in Figure 1e as well as the different LNP behaviour in Figure 4.

The differences in the behavior of the LNPs is an important point raised by the reviewer. Please see the discussion on segregation and dissociation below (2.2 and 2.3) for details.

- 2.2 I would have liked to see some model studies used to attempt to come up with a rationale for the large differences between LNPs containing siRNA and mRNA.

We appreciate the reviewer's encouragement to use model studies to further understand the differences in disintegration behavior between siRNA and mRNA-LNPs, as observed with our imaging data.

A recent paper by Aliakbarinoddehi et al. offers valuable insights that are complementary to our findings, by using a model membrane system to mimic early endosomal conditions. Their findings reveal that siRNA-containing LNPs fully release their cargo upon fusion with the model membrane, with siRNA rapidly dispersing into the bulk solution and the lipid marker spreading across the membrane. In contrast, mRNA-LNPs displayed incomplete disintegration. While a portion of mRNA was released into the bulk solution, a significant fraction of mRNA remained bound to the membrane. These results align with our live-cell

imaging data that demonstrate a more complete disintegration for siRNA-LNPs compared to mRNA-LNPs, and we have added this reference to the Discussion section of the manuscript.

- 2.3 The authors talked about segregation in their discussion of this phenomena but they did not do this experiment with mRNA (only siRNA in Figure 5). It seems like an obvious experiment to improve their study.

The point raised by the reviewer is highly relevant, both for showing similar behavior with respect to segregation between siRNA and mRNA LNPs and possibly compare this property between the two. We have performed additional experiments using BODIPY-MC3 mRNA-LNPs, shown in Figure 5b,c and 6b). In addition, we have expanded our analysis of segregation with a quantitative method to evaluate the colocalization of RNA with both free BODIPY-MC3 and packed (red-shifted) BODIPY-MC3.

In summary, while I think this an important area, I am not convinced that this paper adds a lot of concrete understanding to the area beyond visualising the processes of galectin and ESCRT recruitment correlated with RNA release, thus I would suggest publication only after major revision. I have added some additional points for clarification below.

- 2.5 I would say the LNP internalisation looks similar from 20 nM through to 100 nM (Figure S1a) with the green curve (20nM) tracking higher than the other two. Thus, why is the galectin-9 foci so much lower in the 20 nM case (Figure 1b)?

Thank you for raising this issue. Mainly two factors contribute to a disconnect between the number of intracellular LNP foci and galectin-9 foci at high LNP concentrations. 1) At higher LNP concentrations there is an accumulation of multiple LNPs in individual vesicles, that (at this resolution) are detected as a single “LNP foci” (LNP-containing foci). 2) Crowding of LNP-containing vesicles makes it difficult to detect individual vesicles (foci) (resulting in a “saturation” of the detection). As a result, the number of detectable foci do not fully equate the total number of internalized LNPs (at high doses), while the total number of internalized LNPs drive the production of galectin foci (hence the gradual increase at higher doses)

To address your concerns, we have performed a re-analysis of our imaging data to also show the total fluorescence intensity of RNA-LNPs over time (Figure S1a) which is not sensitive to the counting/saturation issue described above. This measurement of uptake demonstrates dose-dependent LNP uptake that share similar characteristics with the recruitment of galectin-9 in Figure 1b, thereby further clarifying the link between RNA-LNP uptake and galectin-9 recruitment. In addition, we have adjusted the spot counting algorithm (thresholding) to be slightly less affected by the “crowding effect” to better reflect the actual number of LNP spots.

In addition, does the amount of foci correlate to membrane disruption and thus escape.

We have added data showing the correlation between the number of LNP-foci and the number of galectin-9 foci at ~180 min, when cells appear relatively saturated by RNA-LNPs (Figure S1c).

We believe this additional data strengthens the link between LNP uptake and galectin-9 recruitment, providing clearer insights into the dose dependence.

The results on the ESCRT seem to suggest it is as the authors saw 20% increase in foci with ESCRT knockdown. I think the importance of galectin-9 foci could be made clearer?

We appreciate the reviewer's suggestion to clarify the role of galectin-9 and ESCRT recruitment in response to endosomal membrane damage induced by LNPs. ESCRT components respond to smaller membrane perturbations by repairing or sealing damage sites, while galectins are suggested to be recruited to more substantial membrane disruptions, where luminal glycans are exposed. With ESCRT knockdown, small membrane damages that would typically be sealed by the ESCRT machinery remain unrepaired, and this can lead to more extensive damages, as shown by the 20% increase in galectin-9 recruitment. However, we do not yet know the release efficiency of these additional galectin-9⁺ events. We have added text to the Discussion section to clarify this in the revised manuscript.

- 2.6 The authors showed that there was an impact on RNA load for siRNA systems but none for mRNA (line 191-194), why would there be a difference?

We agree that the difference in impact between the siRNA and mRNA systems seems unexpected. However, several factors likely contribute to this discrepancy. The internalization of siRNA-LNPs occurs faster than that of mRNA-LNPs, which may lead to the formation of vesicles with a wider range of payloads (greater extremes). As a result, the difference in load between the lowest and highest siRNA-LNP subgroups (Sup. Fig 4b, or Sup. Fig 2c in initial submission) is nearly 100-fold, compared to a 6-fold difference observed for the mRNA-LNP subgroups. Since RNA load appears to influence release efficiency, one possible explanation is that only a single or a few LNPs release RNA, while the majority remain intact without contributing to productive release. It is conceivable that we might observe lower relative release efficiency with mRNA-LNPs if their uptake led to a higher multiplicity of LNPs per vesicle. However, the more consistent mRNA-LNP payload distribution compared to siRNA-LNPs likely diminishes the differences in relative release efficiency attributable to variations in RNA load.

- 2.7 The paper does not really discuss the intensity of the galectin-9 recruitment which is interesting as it varies quite widely, I think this would be a helpful addition?

Line 195-198 in old manuscript (bottom page 8 of new): "*When subgrouping events with respect to galectin-9 response, there was no difference in the RNA-LNP release kinetics between vesicles showing weak or strong galectin-9 recruitment (Supplementary Fig. 2d).*". (Fig S4c in new manuscript).

This is true for relative release. In absolute quantity the differences are substantial (>10 fold differences when comparing high, medium, low siRNA). This also correlates to gal9 intensity, i.e. more RNA/lipids > more absolute release > more gal9 recruitment (likely bigger damage). In conclusion it is hard to draw simple conclusions on the efficiency of cargo release only from the magnitude or kinetics of gal9 recruitment. The key take home message is that the total relative efficiency is surprisingly stable irrespective of galectin-9 recruitment amount.

- 2.8 I had a question about figure 3a/b, I think the pattern of the recruitment of galectin and EEA1 looks different in the two types of RNA, in siRNA you have lost the signal of EEA1 when the galectin starts to increase but in the mRNA the EEA1 stays consistent long after this time. Is this representative of the rest of the data and if so, what does it mean?

The compartment in figure 3b was mis-labelled and is in fact Rab5 and not EEA1 as stated. We have now changed to the correct compartment labels. Thank you for pointing this out.

- 2.9 Figure 4 clearly shows a change in the siRNA distribution, but I was not sure if these were really single LNPs, the cross-section suggests these are 300-500 nm, this is large?

The FWHM (Full width half maximum) of the upper two line plots of figure 4a (in the old manuscript) are ~210 and ~250 nm respectively (in the multi-LNP vesicle is harder to measure FWHM for individual LNPs). The FWHM of LNPs in EEA1 vesicles in the new Figure 4c and 4d are ~190 and ~200 nm respectively. Resolution of Airyscan at 647 nm is ~180 nm (after Airyscan processing). The observed diameter of LNPs is the convolution of the LNP (80 nm) and the PSF (lateral resolution: 180 nm): ≈ 200 nm (assuming gaussian shaped functions). This is in very good agreement of the multiple LNP size measurements presented in Suppl. Fig. 8 where the modal value of LNP size is at or slightly below 200 nm for both siRNA and mRNA LNPs on both glass slides and in early endosomes. In conclusion we deem the visualized ~200 nm (FWHM) LNPs in early endosomes to be single LNPs.

I would also say the CD63 images do not look like a diffuse breakdown of one or two particles as the fluorescence is not uniform, it still looks punctate in multiple cases (Figure 4b), how do we know this is not the result of more LNPs in the endosomes?

We recognize the reviewer's valid concern regarding the presence of multiple LNPs in late endosomes. While it is plausible that multiple LNPs reside within a single endosome, as suggested by the occasional punctate signal, the more dispersed fluorescence signal observed in late endosomes indicates that many of these LNPs have likely undergone partial or full disintegration. For example, in Figure 4a, we show multiple intact LNPs in an early endosome, where the LNPs appear as distinct foci, clearly more compact than the dispersed signals observed in late endosomes. This shift from compact to dispersed signal as LNPs progress to late endosomes strongly suggest disintegration. To further

substantiate these observations, we have acquired additional data using fast live-cell super-resolution iSIM, where z-stacks of entire endosomes were acquired. This approach further strengthened the differences between early and late endosomes for siRNA-LNP treated cells, as intact LNPs (either single or multiple) are shown as distinct puncta in early endosomes (Figure 4c and d), while the siRNA signal in late endosomes appears more homogeneously dispersed through the entire vesicle (Figure 4e and Supplementary figure 8c-e), further demonstrating the difference between intact and disintegrated LNPs in early and late endosomal compartments.

I also think the same cross-section data should be added for the mRNA data in Figure 4 for comparison.

In response to the reviewer's suggestion, we have also added line profiles for the mRNA-LNPs, which we agree is an important addition for comparison.

2.10 Why wasn't the FRET study done with mRNA and siRNA (Figure 5)? I think this is a limitation with the manuscript as everything to this point has been done with point and they show clear differences, this experiment might have offered some explanation for these findings?

We appreciate the reviewer's suggestion to study BODIPY-MC3 LNPs containing either siRNA or mRNA, as it indeed strengthens the mechanistic insights into the differences between these two types of cargos. In response, we have conducted additional experiments, now included in the revised manuscript. Specifically, we have investigated the subcellular localization of intact and disintegrated BODIPY-MC3 LNPs loaded with siRNA or mRNA using the imaging approach already described in the manuscript, as well as with a newly developed quantitative analysis. In short, we quantified the spatial distribution of free BODIPY-MC3⁺ (disintegrated), and FRET⁺ (intact) LNPs, as well as redshifted BODIPY-MC3⁺ LNPs (dense) relative to the nucleus and normalized to the outermost foci within each cell. The analysis demonstrated that disintegrated LNPs were more frequently localized in perinuclear structures (i.e. later endolysosomes) compared to intact LNPs, and that this was similar for siRNA and mRNA containing LNPs (Fig. 5b, c). However, we observed that redshifted MC3⁺ LNPs, reflecting LNPs with dense lipids, were more commonly found in perinuclear regions of LNP-mRNA treated cells, suggesting that endosomes containing aggregated ionizable lipid are more prevalent in late endosomes for mRNA-LNPs, consistent with the mRNA-cargo aggregates observed in late endosomes (Fig. 4d).

Additionally, we expanded our analysis of individual endosomes containing BODIPY-MC3 in the membrane to include both siRNA- and mRNA-LNPs (previously only performed for siRNA-LNPs). Our new data confirm that in a subset of endosomes, BODIPY-MC3 associates with the endosomal membrane. Furthermore, nanodomains with redshifted BODIPY-MC3, showing local enrichment of ionizable lipid, were observed to localize in

direct proximity of membrane-tethered RNA remnants, for both siRNA- and mRNA-LNPs (Fig. 6c and Supplemental Figure 9f).

I also think the evidence of separate BODIPY and siRNA is pretty weak (I think it is only demonstrated using one small image in Figure 5b). I don't find this convincing.

We agree with the reviewer that this point could be clarified, and that further data would aid in that. Based on our observation that endosomes containing free BODIPY-MC3 were more prevalent in perinuclear regions compared to those with dense (redshifted) BODIPY-MC3, we developed a quantitative analysis to measure the colocalization of RNA⁺ endosomes with either free or dense BODIPY-MC3. We showed that endosomes with dense BODIPY-MC3 were more likely to contain RNA compared to those with free BODIPY-MC3 (Fig. 6a, b). This indicates that RNA⁻ endosomes are more frequent in perinuclear regions and provides further evidence that RNA cargo and ionizable lipid can separate during endosomal sorting. However, the reported fraction of RNA⁺ endosomes is likely underestimated due to sensitivity limitations of the microscopy assay and potential channel misalignment caused by time delays during filter changes.

Collectively, we have repeated all key experiments for siRNA-containing BODIPY-LNPs using mRNA-containing BODIPY-LNPs. Additionally, we have included quantitative analyses to further demonstrate that LNP disintegration as well as segregation of RNA cargo and ionizable lipid occur during endosomal maturation, for both LNPs containing siRNA and mRNA.

The authors claim that the addition of BODIPY into the membrane supports it plays a role in membrane disruption (line 360-361), while I think that is likely true I do not think this data proves it. Can the authors please explain this point more clearly?

We show that both membrane damage markers and high local concentrations of BODIPY-MC3 in endosomal membrane nanodomains localize in direct proximity to membrane-tethered siRNA-remnants (Fig. 6c and Fig. 7a-e), strongly suggesting that high concentration of ionizable lipid is associated with localized membrane damage.

However, we agree that further data would clarify our rationale and strengthen this claim. Therefore, we have included new imaging data in the revised manuscript, examining the sub-endosomal localization of BODIPY-MC3 in galectin-9 marked vesicles. In these experiments, cells expressing mCherry-galectin-9 were treated with AF647-siRNA BODIPY-MC3 LNPs, and we assessed endosomes with membrane associated BODIPY-MC3 and galectin-9 recruitment. Our analysis showed that galectin-9 was frequently recruited to endosomal membrane nanodomains enriched with BODIPY-MC3 and membrane-tethered siRNA (new Figure 6f). This additional data provides more insight into how MC3 localization at the membrane correlates with sites of endosomal membrane damage. We believe this strengthens our findings and addresses the reviewer's concern about the relationship between BODIPY-MC3 and endosomal membrane damage.

Reviewer #3 (Remarks to the Author):

In the manuscript titled “Cellular and biophysical barriers to lipid nanoparticle mediated delivery of RNA to the cytosol” authors use microscopy to unravel various processes involved post the endocytosis of RNA-LNPs. By tracking single vesicles containing LNPs, the authors describe the damage marked by galectin recruitment. Further, the co-localization of LNPs with different endosomal compartments (EEA1, Rab5, CD63) is quantified and the data shows the highest load of RNA in early endosome vesicles marked by Rab5. Using super-resolution microscopy, the authors image the disintegration of LNPs to be happening in late endosome vesicles marked by CD63. Finally, the recruitment of gal9 to Rab5/EEA-1 vesicles containing siRNA-LNPs is microscopically evaluated to determine the points of damage to the endosomal membrane. Overall, the study employs various microscopic tools to gain mechanistic insight into the process of endosomal escape. The observations reported are important to the burgeoning field of LNP-based therapeutics. While the manuscript is meticulously written and microscopic data is detailed and rigorous, there are some concerns with the interpretation and robustness of the data. Insight into the bigger picture and how the observations might affect the choice of the vehicle and the payload is missing.

- 3.1 In Figure 1f, the decrease in the fluorescence intensity upon Gal9 recruitment is interpreted as an event of endosomal release of the RNA. Why is the drop in fluorescence interpreted as the release of mRNA?

This is a fundamental and important point raised by the reviewer. There are several aspects that support that the observations we report represent release of RNA payload.

- 1) Gal9 recruitment requires exposure of luminal endosomal glycans to the surrounding cytoplasm. This is caused by a damage to the vesicle lipid bilayer and galectins are widely used to detect membrane damage. We are not aware of any study showing that such glycans can be exposed to the cytosol and trigger galectin recruitment without the need of a lipid bilayer disruption of some nature.
- 2) Gal9 recruitment caused by a range of membrane-damaging substances and agents have been found to mark events where material or microbes escape the endosomal compartment into the cytoplasm.
- 3) Specifically, we have previously observed galectin recruitment (on a single vesicle or particle level) for dextran (Du Rietz et al, Nat Comm 2020) and cationic siRNA lipoplexes (Wittrup et al Nat Biotech 2015, Hedlund et al, Nat Comm 2023) where galectin+ damages and concomitant drops in vesicle fluorescence was associated with cytosolic signal of the cargo as evidence of endosomal escape.
- 4) Here, vesicles releasing RNA payload maintain a lower fluorescence intensity after galectin recruitment (post-release), excluding the possibility of technical artifacts due to e.g. particle tracking (suboptimal fitting of the object mask).
- 5) A comprehensive strategy has been used for addressing photobleaching, and the sudden decrease in the intensity as observed is not indicative of fluorophore degradation.

- 6) Other possible biological reasons for a sudden decrease of the intensity (fission of vesicles, neighboring vesicles suddenly separating etc) occur and but are associated with detectable particles separating and such events are excluded from quantifications.

Have the authors considered the possibility of RNA being degraded inside the vesicles?

RNA degradation is likely to occur as the payload reaches later endosomal compartment. The fluorescence intensity of the AF647 and Cy5 fluorophores that are detected are relatively insensitive to pH and have a high stability even in degradative endosomal compartments (as compared to e.g. protein based fluorophores). Thus, RNA degradation will not per se cause loss of fluorescence intensity mimicking the release behavior that we observe. Secondly, the kinetics (intensity drop during a few second until stability) is not typical of degradative processes.

Additionally, in case there is an actual endosomal escape happening, the decrease in the fluorescence inside the vesicles will lead to a corresponding increase in the fluorescence outside the vesicles (i.e., cytoplasm); can the authors observe this event?

We agree that a decrease in fluorescence within vesicles due to endosomal escape would in principle correspond to an increase in fluorescence in the cytoplasm (and precisely this has been shown for larger lipoplexes e.g. Hedlund et al. Nat Comm 2023). However, detecting such an event for the *much* smaller LNPs is challenging for several reasons.

First, the fluorescence intensity in the cytoplasm is likely below the detection threshold of our imaging system. This is because, at most, only about 50% of damaged vesicles result in productive release, and these events occur intermittently, delivering only a fraction of the RNA load. Additionally, the released RNA becomes significantly diluted within the substantially larger cytoplasmic volume compared to its concentrated state within vesicles, making the fluorescent signal weak and difficult to detect reliably.

Second, any fluorescence increase in the local cytoplasm is likely to be extremely transient and difficult to capture for such weak release events. The released RNA will diffuse away rapidly, further reducing the likelihood of detecting a measurable signal.

While it would indeed be valuable to measure shifts in fluorescence between vesicular and cytoplasmic compartments using fluorescently labeled RNA, for LNPs such an investigation would likely require a single-molecule detection strategy to capture these subtle and transient events with sufficient sensitivity.

- 3.2 Throughout the manuscript, information should be provided on the number of cells being analyzed for each data set presented in the graphs.

We agree that including the number of cells analyzed for each dataset is essential for

transparency. We have now provided this information throughout the manuscript for each dataset presented in the graphs, as seen in the updated figure legends.

- 3.3 Fig1e: Do the authors have a hypothesis on why only a fraction of gal9+ vesicles contain the mRNA? What was the time point for data analysis in Fig 1e post-LNP addition? Did this fraction change with time post-LNP addition?

In the discussion, we hypothesize that the lower hit rate for mRNA-LNPs may result from the segregation of ionizable lipids and RNA into distinct endosomal compartments, leading to LNP-induced disruptions in vesicles with little or no RNA payload. Differences in lipid composition and RNA binding could influence LNP stability and compartmental segregation, contributing to this variability. The alternative hypothesis that most vesicles have already released their RNA cargo is unlikely, given the strong correlation between galectin recruitment and cytosolic siRNA release, and the lack of release from ESCRT-positive perturbations. Partial exocytosis of RNA cargo may also play a role.

Image acquisition was typically initiated when multiple galectin-9 foci appeared, generally between 0.5–2.5 hours after LNP addition. During this time frame, the proportion of damaged RNA+ to RNA- vesicles remained fairly constant, both during image acquisition and across different time points after LNP addition, for both siRNA and mRNA. This has been clarified in the new Supplementary Figure 3.

- 3.4 In Fig2, the authors report the no. of foci of CHMP2A analyzed. It is important to also provide information on the no. of cells being evaluated. In addition to this, it is more informative to the readers if images of more than one cell are provided in the main figures.

We have added the number of cells evaluated, alongside the reported number of CHMP2A foci, providing a clearer context for analysis. Additionally, we have added a new figure (Fig. 2A), showing multiple representative cells, as recommended. We agree that these changes offer a more representative view of the CHMP recruitment. Still, zoomed out images of CHMP recruitment are somewhat difficult to interpret and appreciate as the intensity of CHMP foci are generally quite low (very dim compared to galectin-foci) and primarily analyzable in medium expressing cells only. We have included both zoomed in and zoomed out views and also time-course snaps to convey the data as fully as possible.

- 3.5 On pg10, line 256, the authors write “EEA1 intensity generally peaked before endosome damage, while Rab5 was more consistent over the observed interval”. It will be good to discuss why this is happening and if it relates to the signature of the compartment where the endosomal release is happening.

This observation is reminiscent of lipoplex release which seem to proceed from EEA1 negative vesicles directly after having been positive (Wittrup et al. Nat Biotech 2015). Here we see also many damage events in vesicles still positive for EEA1 indicating a possibly slightly earlier damage window. In summary we interpret the data as damage to early endosomes or endosomes on its path to become maturing endosomes and starting to

lose early endosome markers (i.e. EEA1). The results section has been updated with this clarification.

- 3.6 Following this, if the endosomal release is occurring from Rab5/EEA1[±] vesicles, why is the RNA load similar in all the endosomal compartments (supp fig 4b)?

We interpret the reviewer's question as suggesting that differences in RNA load might be expected between endosomal compartments, potentially due to variations in the extent of vesicle damage across these compartments. However, the RNA load in Figure S4b/ Figure S6d (initial-/current version), was measured prior to the onset of detectable endosomal damage indicated by galectin-9 recruitment. Therefore, we would not expect any significant variation in RNA load, unless it resulted from other forms of vesicle damage that do not recruit galectins. Nevertheless, our data suggests that such alternative forms of damage would be unlikely to have a substantial impact on RNA load, as the levels observed were consistent across compartments.

- 3.7 Further, with the increase in the time of incubation of mRNA/siRNA LNPs with the cells, is there a change in the proportion/distribution of endosomal compartments-mRNA?

Yes, we observe a clear shift in LNP localization between early and late endosomal compartments with prolonged incubation of siRNA-LNPs, which we have summarized in Supplementary Figure 6a. Cells expressing fluorescently labeled early endosomal marker (Rab5) and late endosomal marker (CD63) were treated with siRNA-LNPs and imaged for an 8 hour period with a confocal microscope. Quantitative image analysis revealed that during the initial hour of incubation, a fraction of the LNPs are localized in early endosomal compartments (Rab5). A significant fraction is then localized with late endosomal compartments (CD63), where they remained during the rest of the observation (8 hours). These results suggest that internalized LNPs are sorted as typical endosomal cargo, progressing from early to late endosomes to later endosomes in line with canonical trafficking pathways.

- 3.8 Colocalization of gal9, EEA1 and RNA is an (fig 3A) observation. More proof is required to prove that Rab5 is the compartment where endosomal release occurs. Will the endosome escape (mRNA expression) be affected if you knock down/downregulate Rab5?

We agree with the reviewers that, while the observation in Figure 3a indicate that Rab5⁺ EEA[±] is the releasing compartment, additional quantitative data on the release from these compartments would support our hypothesis. To address this, we now provide release kinetics data obtained from our compartment-specific analysis, included as a new Supplementary Figure 7. For mRNA-LNP, our findings indicate that Rab5⁺ compartments are a primary site for rapid release, accounting for an overwhelming fraction of damaged vesicles showing fast release kinetics. Similarly, a smaller but consistent fraction of EEA1⁺ structures also contribute to RNA escape, further supporting the role of a Rab5⁺ EEA[±] signature for releasing vesicles. For siRNA-LNPs, we observe rapid release events in both Rab5⁺ and EEA1⁺ vesicles but a substantial fraction of events

are also negative for these markers when imaged with one marker expressed at the time. Taken together, for both mRNA and siRNA LNPs the dominant compartment of release is Rab5⁺ EEA^{+/-} early endosomes but for siRNA cargos other vesicles can also contribute. Figures and text have been updated.

- 3.9 Based on data presented in Fig4., the disintegration of LNPs occurs in the late endosome stages and not in the early stages as depicted in the microscopic images. Since the mRNA release can occur only after the LNPs start to disperse, one would expect the release to occur from CD63⁺ compartments/stages later than Rab5⁺ compartments. It would be great if the authors could shed some light on this and correlate this data with the observations in the previous figure.

We appreciate the reviewer's observation regarding the timing and location of LNP disintegration. To clarify our data, in Figure 4, we show that in EEA1⁺ early endosomes, the majority of LNPs remain intact, while in CD63⁺ late endosomes, most of the LNPs appear disintegrated. However, we also observe that Rab5⁺ early endosomes contain a range of siRNA-LNPs with varying morphologies, including well-defined, intact LNPs, as well as vesicles with partially condensed or more homogenous siRNA distributions. This latter LNP appearance is similar to the disintegrated LNPs observed in CD63⁺ late endosomes. These observations suggest that the LNP disintegration proceeds in Rab5⁺ early endosomes, rather than being confined to the later endosomal compartments. To clarify this point, we have added a sentence in the Results section, to highlight that we believe the LNP disintegration takes place primarily in Rab5⁺ endosomes.

- 3.10 Why have the authors used ApoE in the cell culture media? Since the media is supplemented with FBS, the proteins present in the serum will enable the endocytosis of LNPs.

We include ApoE in the culture media to enhance LNP uptake and this is widely used in the field for in vitro studies of LNPs. Human serum can also be used but in our hands, this led to generally slower kinetics making especially our imaging experiments slower and less efficient. Thus, we have used ApoE to generate a sufficient number of endocytosed LNPs and damage events to obtain a robust sample size for statistical analysis.

In Fig3: Since the mRNA expression/siRNA activity increases with increasing concentration of RNA, it will be good to see if increasing RNA conc. in the medium leads to an increase in the number of different endosomal compartment vesicles.

We agree that this is important to verify to ensure that the compartment analysis is not influenced by any unintended effects caused by the LNPs. We have added new imaging data that address this question in Supplementary Figure 6b. HeLa cells expressing fluorescently labeled early endosomal marker Rab5 or late endosomal marker CD63 were treated with a low dose (10 nM siRNA-LNP), same dose as compartment analysis experiment (50 nM siRNA-LNP), or no LNPs (control) and imaged with a confocal microscope over an 8 hour period, and the number of compartment marker foci were then

quantified. We could conclude no apparent formation of new compartment vesicles (foci) of neither early nor late endosomes, with similar foci counts per cell between all conditions. Thus, we do not believe we induce or consume early or late endosomes during LNP incubation.

- 3.11 Pg14 lines 347 and 346, it would be informative to show the colocalization of Bodipy-MC3 LNPs with gal 9 vesicles.

We have added new imaging data to address colocalization of BODIPY-MC3 LNPs with galectin-9 vesicles (new Fig. 6f). More precisely, we treated cells expressing mCherry-galectin-9 with AF647-siRNA BODIPY-MC3 LNPs and examined endosomes with membrane-bound BODIPY-MC3 and galectin-9 recruitment. Our analysis showed that galectin-9 was frequently recruited to endosomal membrane nanodomains where high concentrations of BODIPY-MC3 and membrane-tethered siRNA localized. We believe this additional data improves the manuscript by providing more insight into the interaction between MC3 localization and endosomal membrane damage.

- 3.12 In suppl. Fig7., the label on the right panel of the image should be corrected to CD63+ vesicles. Interestingly, in this panel, only a few CD63+ vesicles appear to contain the dispersed RNA and the LNPs outside CD63+ vesicles appear diffused/disintegrated. At this stage, are the LNPs present in some other vesicle types eg., those marked by Rab7/Lamp1?

We thank the reviewer for their attention to detail and we have corrected the headings. Furthermore, we have separated the channels of the images to provide a clearer view of the colocalization between channels. As shown in the new figure (Supplementary figure 10 a and b), several of the dispersed siRNA structures are indeed CD63⁺ endosomes, although this colocalization was not apparent in the merged image of the original manuscript, due to very high siRNA signal. However, we also observe dispersed LNP structures that do not colocalize with CD63⁺ endosomes. Based on these observations, we hypothesize that these LNPs are present in other compartments, particularly LAMP1⁺ lysosomes, as suggested by the reviewer.

- 3.13 In Fig. 4, confocal microscopy is used to determine the localization of siRNA within different endosomal compartments. Although ingenious and interesting, this method does not enable, without a z-stack, to appreciate siRNA distribution in the entirety of the vesicle, and thus, to completely trust the resulting observations.

In response to the reviewer's concern regarding the limitations of 2D confocal microscopy for fully assessing the siRNA distribution within vesicles, we have added new data obtained using fast live-cell 3D super-resolution VT-iSIM (Figure 4b and Supplementary figure 8c-e). In this analysis, we have acquired z-stacks of entire endosomes to provide a comprehensive view of the siRNA distribution within the vesicles. These new data confirm our initial findings, with LNPs appearing as distinct foci in early endosomes, while LNPs in late endosomes display homogeneously dispersed signal within the entire vesicle in late

endosomes, indicating substantial disintegration of LNP-siRNA during endosomal maturation.

- 3.14 In Fig. 2b, an increase of CHMP2A foci in the presence of RNA cargo is reported, suggesting the recruitment of membrane repair systems. Considering the images displayed in Fig. 2. a, it would be interesting to exploit them further to quantify the colocalization of such foci with fluorescent RNA precisely.

We appreciate the reviewer's suggestion and we have conducted additional analysis to quantify the colocalization of CHMP2A foci with RNA. The CHMP2A foci identified at 1 and 3 h after treatment were manually inspected and classified as either positive or negative for AF647-siRNA or Cy5-mRNA signal (Fig. 2d).

- 3.15 Finally, authors use MC3 as an ionizable lipid for all their studies. This is not the best option for mRNA delivery. It will be interesting to see how the observations change with the change of ionizable lipids for example with ALC-0315 (Pfizer/bnt) or SM-102 of Moderna.

We recognize that other ionizable lipids, such as ALC-0315 (used in the Pfizer/BioNTech formulation) or SM-102 (used by Moderna), are more optimized for mRNA delivery. We agree it will indeed be interesting to see whether different ionizable lipids influence the observed LNP behavior and membrane interactions. In response to this important point, we have added text to the Discussion section, to acknowledge this limitation and suggest that future studies could use methods presented in this manuscript to investigate how alternative ionizable lipids might impact LNP disintegration, intracellular trafficking and RNA release.